# Immune-associated molecular classification and prognosis signature of sepsis

**Zhiwei Li**[1⊕], **Leyi Wang**[2⊕], **Shuting Yang**[1], **Bin Luo**[1], **Yezi Liu**[1], **Mengsi Chen**[1], **Changmin Wang**[1]*

1 Clinical Laboratory Center, People's Hospital of Xinjiang Uygur Autonomous Region, Urumchi, Xinjiang, China, 2 Department of Nursing, People's Hospital of Xinjiang Uygur Autonomous Region, Urumchi, Xinjiang, China

⊕ These authors are contributed equally to this work.
* W13609992616@163.com

## Abstract

This study aims to explore the molecular subtypes of sepsis and the correlation between immune-related genes and the prognosis of patients with sepsis. Utilizing the Gene Expression Omnibus dataset (GSE65682) with 479 patients with sepsis as the training set and 164 patients treated at our hospital as the independent validation cohort. An unsupervised cluster analysis was used to identify potential molecular subtypes of sepsis, and a weighted gene co-expression network analysis was performed to identify gene modules. Gene Ontology, Kyoto Encyclopedia of Genes, and Genomes enrichment analyses were performed, and the immune status was also evaluated. Using LASSO regression and multivariate Cox regression, an immune-related gene prognostic model was developed, validated, and evaluated, followed by an individual risk scoring system. We identified two molecular subtypes of sepsis that are associated with distinct immune response patterns and clinical outcomes. Patients in Cluster A exhibited poorer survival and enrichment of pro-inflammatory pathways, while those in Cluster B had better outcomes and enrichment of immune regulatory pathways. A 10-gene prognostic model was constructed, stratifying patients into high- and low-risk groups using the estimated risk score that was confirmed to be an independent prognostic factor in both the training (hazard ratio [HR]: 1.126, 95% confidence interval [CI]: 1.096–1.156, P < 0.001) and validation datasets (HR: 1.149, 95% CI: 1.085–1.216, P < 0.001). A risk scoring system was developed based on the risk score and clinical parameters, with estimated mortality probabilities of 0.132 (7-day), 0.211 (14-day), and 0.258 (21-day). High-risk patients had significantly worse prognoses, and this was validated in the independent cohort. Distinct immune cell profiles were found between the two subtypes and risk groups, with B cells, CD8 + T cells, and NK cells elevated in Cluster B. This study identified immune-related molecular subtypes of sepsis and developed a prognostic model that accurately predicts sepsis mortality. These findings provide insights into the immune

**Data availability statement:** The expression levels of immune-related genes from sepsis patients can be available from the GEO without any special privileges (GSE65682: https://www.ncbi.nlm.nih.gov/geo/query/acc.cgi?acc=GSE65682), which is publicly for everyone. The validation dataset and original codes are within the manuscript and its Supporting Information files.

**Funding:** The author(s) received no specific funding for this work.

**Competing interests:** The authors have declared that no competing interests exist.

dysregulation in sepsis and can potentially be used for developing personalized treatment strategies and improving clinical decision-making in sepsis management.

## Introduction

Sepsis is a life-threatening condition characterized by organ dysfunction resulting from a dysregulated host response to infection [1,2]. It is conservatively estimated to be one of the leading causes of death and critical illness worldwide [3]. Despite advances in sepsis treatment over the past few decades, the mortality rate for septic shock remains alarmingly high, ranging from 25–30%, and in some cases, reaching 40–50% [4]. Furthermore, survivors of sepsis often suffer from long-term physical, psychological, and cognitive impairments [5,6]. Unlike other major diseases, sepsis lacks specific treatment, with no approved drugs available, and its treatment relies instead on supportive care, antibiotics, and hemodynamic stabilization [7]. This highlights the urgent need to identify novel diagnostic biomarkers and develop prognostic models to enhance clinical decision-making.

Immunodysregulation plays a central role in sepsis. Monocytes and macrophages, key components of the innate immune system, are critical in orchestrating the host's immune response during sepsis. In its early stages, the excessive secretion of pro-inflammatory cytokines and chemokines by two distinct lymphocyte populations exacerbates the inflammatory response, thereby increasing mortality [8,9]. Neutrophils, the most abundant white blood cells in peripheral circulation, have been shown to contribute to immunodysregulation in patients with sepsis, leading to multi-organ failure through the release of cytokines and reactive oxygen species. CD4+T cells play a direct role in modulating the host's response to sepsis, and sepsis-induced apoptosis can result in CD4+T cell exhaustion [10,11]. Additionally, sepsis leads to long-term changes in the phenotype, function, and localization of memory CD8+T cells, which may impair the host's ability to respond to reinfection [12]. Therefore, immune-related genes hold great promise for improving the diagnosis and prognosis of sepsis.

Bioinformatics approaches have been extensively employed to identify key gene clusters associated with human disease progression, offering significant insights into potential molecular mechanisms and aiding the discovery of clinical diagnostic and therapeutic targets [13]. The pathogenesis of sepsis is highly complex, involving factors such as inflammation, immune dysregulation, and coagulation disorders [14]. Previous studies have focused on the effect of individual genes on sepsis. Arunachalam et al. investigated the role of P2Y2 purinergic receptors in liver injury and sepsis. They found that mice lacking P2Y2 receptors exhibited reduced inflammation, less liver damage, and improved survival in response to both inflammatory liver injury and sepsis. These P2Y2 knockout mice also maintained normal serum arginine levels, preventing immune dysregulation and increased bacteremia seen in wild-type mice. The findings suggest that P2Y2 receptors play a crucial role in the pathophysiology of liver injury and sepsis, and targeting this receptor could potentially mitigate cytokine

storms and improve outcomes in sepsis [15]. Liu et al. found that FCGR2C was the only down-regulated gene differentially expressed between survivors and non-survivors in a cohort of 81 septic patients. FCGR2C was more predictive than the SOFA score in evaluating the prognosis of septic patients [16]. Jian et al. found that the level of UCP2 in blood cells of sepsis patients was significantly higher than in healthy controls, both at the mRNA and protein levels. UCP2 in blood cells may serve as a specific biomarker for sepsis, and its level is positively correlated with the severity of sepsis [17]. These studies on individual genes have been insufficient in fully elucidating the complex pathophysiological processes of sepsis. There is also a study that reported the roles of immune-related genes in the prognosis of sepsis [18]. However, the previous study lacked comprehensive analysis of data from multiple perspectives, such as molecular subtypes, immune-cell infiltration, and individual risk evaluation. Additionally, the model's training performance had some limitations. More importantly, previous studies have not included valid independent cohort validation data. Therefore, it is essential to conduct a more thorough analysis and further validation of prior findings. To address these issues, we obtained the peripheral blood gene expression profiles of patients with sepsis with varying survival outcomes from the Gene Expression Omnibus (GEO) database. By focusing on immune-related genes linked to sepsis prognosis, we explored potential molecular subtypes of sepsis and compared their prognostic characteristics and functional differences. Through a weighted gene co-expression network analysis (WGCNA), we predicted the key genes that are associated with these sepsis subtypes and performed Gene Ontology (GO) enrichment, Kyoto Encyclopedia of Genes and Genomes (KEGG) enrichment, immune infiltration, and survival analyses. Based on the immune-related genes, we then constructed and evaluated a prognostic model to predict mortality in patients with sepsis. Additionally, we validated this model using our independent cohort data. Moreover, a personalized mortality risk assessment system was developed based on the identified model genes. This study provides novel insights into the core genes and molecular mechanisms involved in sepsis prognosis, offering promising avenues for developing treatment strategies and prognostic risk management.

## Materials and methods

This study followed the Tripod checklist prediction model development and validation (S1 File).

### Patients and samples

We downloaded the blood transcriptome dataset (GSE65682) of patients with sepsis from the GEO public database (http://www.ncbi.nlm.nih.gov/geo/). This dataset includes data from 802 patients with sepsis, of whom 479 had available data on clinical characteristics (e.g., survival status, survival time, age, sex, pneumonia type, thrombocytopenia, ICU-acquired infection type, diabetes, and endotype) and follow-up. These were used as the training dataset. According to the Sepsis-3 criteria: The diagnosis of sepsis is based on the SOFA (Sequential Organ Failure Assessment) score, which assesses the extent of organ dysfunction. A ≥ 2 increase in SOFA score indicates sepsis [19]. Additionally, we collected data from 164 patients with sepsis treated in our hospital between January 7th, 2024 and July 30th, 2024. This served as an independent validation dataset. Blood samples and clinical information (age, sex, pneumonia, thrombocytopenia, ICU-acquired infection type, diabetes, and APACHE II score) were collected. We also recorded 28-day follow-up outcomes (primary outcome: mortality) and the survival duration for further analyses. Sepsis mortality, age, and gender were evenly distributed across both datasets. However, the training dataset had some missing entries, and significant differences in the distribution of pneumonia, thrombocytopenia, infection type, and diabetes were noted between the training and validation sets. All original data were provided in S2 File, and original code was presented in S3 File. The immune-related genes were sourced from the Immunology Database and Analysis Portal (https://www.immport.org/), which includes 2,498 genes that were previously reported in a study, and was also provided in S4 Table [18]. This study was approved by the Ethics Committee of People's Hospital of Xinjiang Uygur Autonomous Region. Written consent was obtained from all participants, and data was analyzed anonymously.

## Identification of sepsis molecular classification

First, we identified prognosis-related genes from the pool of immune genes. We then performed unsupervised cluster analysis using the "ConsensusClusterPlus" package in R to identify potential molecular subtypes. Before the cluster analysis, we first excluded the genes with a median absolute deviation of < 0.5 [20]. Non-negative matrix factorization clustering was performed, and the cophenetic correlation coefficients and cumulative distribution function were calculated to determine the optimal k value. The consensus matrix legend was used to determine the optimal k-value, while empirical cumulative distribution function plots were used to display the consensus distributions for each k value [21]. The K value will be selected when the cophenetic correlation coefficient is higher with flatter cumulative distribution function and significant differences in prognosis. We validated the molecular subtypes through a principal component analysis (PCA) and t-distributed stochastic neighbor embedding (t-SNE).

## Gene set variation analysis

The gene set variation analysis (GSVA) was conducted using the R package "GSVA," a non-parametric, unsupervised method designed to assess gene set enrichment in microarray and RNA-seq data [22]. GSVA converts the gene expression matrix into a gene set expression matrix using the MSigDB database (c2.cp.kegg.v7.4.symbols) to evaluate the enrichment of different metabolic pathways across samples. Subsequently, the differential analysis was performed between different molecular subtypes, and a log 2-fold change > 0.2 and P < 0.05 were considered significant.

## Identification of differentially expressed genes

We identified differentially expressed genes (DEGs) between molecular subtypes or risk groups using the "limma" package (version 3.46.0) [23]. DEGs were defined as genes with an adjusted p-value of less than 0.05 (Bonferroni method) and an absolute log 2-fold change greater than 0.1. A volcano plot was created to visualize the results.

## GO enrichment and KEGG pathway analyses

GO was used to classify gene functions into biological processes, cellular components, and molecular functions [24]. GO enrichment analysis was performed using the "clusterProfile" package in R. Additionally, Kyoto Encyclopedia of Genes and Genomes (KEGG) pathway enrichment analysis was conducted with the "GOplot" package to explore pathway involvement [25]. For enrichment analyses, the adjusted P values were adopted using the Bonferroni method.

## Weighted gene co-expression network analysis

Hierarchical clustering of the gene expression matrix was performed to identify and exclude outlier samples. We then conducted a weighted gene co-expression network analysis (WGCNA) to identify co-expressed gene modules [26]. The optimal soft-thresholding power was determined based on scale-free topology fit. When constructing a gene co-expression network, in order to make the network resemble a scale-free network, an appropriate β value, also known as the soft threshold parameter, must be selected. This value determines how the connection weights in the network increase as the correlation between genes strengthens. Generally, the goal in selecting this value is to make the relationship between the network's average connectivity (k) and the logarithm of the number of genes (log(k)) appear linear on a log-log plot. It is also desirable for the degree distribution of the network to exhibit scale-free distribution characteristics. To select the optimal soft threshold, the scale-free topology fit index under different β values can be observed. When this index approaches or reaches 0.9, it indicates that the β value is appropriate. A dissimilarity matrix was calculated from the topological overlap matrix for hierarchical clustering. Modules with at least 30 genes were identified using dynamic tree cutting, and module eigengenes were calculated to assess module-module relationships. Modules with a dissimilarity of less than 0.25 were merged, and a heatmap was used to visualize the module relationships.

## Protein-protein interaction network

Key genes were identified by intersecting DEGs with WGCNA module genes. We used the STRING (https://cn.string-db.org/) database to build the protein-protein interaction, and interaction correlations were calculated and input into Cytoscape (Version 3.9.1) to construct a protein-protein interaction network with an interaction score of > 0.4. Using the degree algorithm of the hub module, the top 10 hub genes associated with sepsis subtypes were identified.

## Immune infiltration analysis

We utilized CIBERSORT, a deconvolution tool that uses linear support vector regression, to estimate the relative abundance of human immune cell subtypes (T cells, CD8 + T cells, aDCs, B cells, DCs, iDCs, macrophages, neutrophils, natural killer [NK] cells, pDCs, T helper cells, Tfh, Th1, Th2, tumor-infiltrating lymphocytes [TIL], regulatory T cells, central memory T cells, APC co-inhibition and co-stimulation, CCR, check-point, cytolytic activity, HLA, MHC class I, parainflammation, T cell co-inhibition and co-stimulation, and Type I and II IFN response), particularly in complex cell mixtures derived from microarray or sequencing data [27]. The CIBERSORT package (0.1.0) was employed to assess immune cell infiltration across samples. Box plots were used to visualize differential immune expressions across subtypes and risk groups. Spearman correlation coefficients were calculated to examine the relationship between risk scores and immune cell infiltration. Key immune gene expression differences between molecular subtypes and risk groups were also compared.

## Establishment and validation of the immune-related gene prognostic model for sepsis

To construct the immune-related gene prediction model for sepsis, we first applied least absolute shrinkage and selection operator (LASSO) regression to the prognostic genes identified from the univariate Cox regression analysis. A multivariate Cox regression analysis was subsequently performed on the genes selected by LASSO. A risk score for each patient was calculated using the regression coefficients and gene expression levels: risk score = $\beta 1 \times geneExp1 + ... + \beta (N) \times geneExp (N)$. Patients were stratified into high- and low-risk groups based on the median risk score, and Kaplan–Meier survival curves were plotted. The prediction model was validated using our independent cohort dataset.

## Independent prognosis and risk scoring system

To assess the independent prognostic significance of the risk score, univariate and multivariate Cox regression analyses were conducted. Potential confounding factors, including age, sex, pneumonia type, thrombocytopenia, ICU-acquired infection type, diabetes, and endotype, were adjusted in the multivariate analysis. Some missing data was included for pneumonia type, thrombocytopenia, infection type, and diabetes in the training dataset. To make full use of this data, we handle the missing data as the third type of variable. Univariate and multivariate Cox regression analyses were also performed on our validation cohort using different clinical parameters (age, sex, pneumonia, thrombocytopenia, ICU-acquired infection type, diabetes, and APACHE II score). A nomogram-based risk scoring system was developed to estimate individual mortality risk based on model genes and clinical parameters. The model's predictive performance was evaluated using a receiver operating characteristic (ROC) curve, while calibration plots assessed the model's fit at 7, 14, and 21-day intervals. A power is calculated to ensure sample size sufficiency using Power Analysis and Sample Size 15.0.

## Quantitative reverse transcriptase polymerase chain reaction

Total RNAs from ten genes (CAMP, CETP, CXCR3, DDX17, DEFA4, DEFB119, FASLG, FGFR1, HLA-E, HSPA1B, IL8, MAP3K14, NFYC, NOX4, NRG1, SDC4, SOS1, TNFRSF12A, TNFRSF14, TNFSF12, and UNC93B1) were extracted using Trizol reagent (Invitrogen, USA) following the manufacturer's protocol. The primer sequences are provided in S5 Table. RNA was quantified and assessed for purity using a NanoDrop 2000 spectrophotometer, and its integrity was

checked by 1% agarose gel electrophoresis. One microgram of total RNA was reverse transcribed into cDNA using the RevertAid First Strand cDNA Synthesis Kit (Thermo Fisher Scientific, USA). Quantitative polymerase chain reaction (qPCR) was performed using a 7500 Fast Real-Time PCR System (Applied Biosystems, USA) in a 20 μL reaction containing 10 μL SYBR Green PCR Master Mix, 0.8 μM primers, 2 μL cDNA, and 6.4 μL RNase-free water. The thermal cycling conditions were 95°C for 30 seconds, followed by 40 cycles of 95°C for 5 seconds and 60°C for 30 seconds. GAPDH was used as the endogenous control, and relative gene expression was analyzed using the $2^{-\Delta\Delta Ct}$ method.

## Results

### Identification of sepsis molecular subtypes

The study's flowchart is shown in Fig 1. We obtained 479 records from 890 samples after excluding some without follow-up information and MAD < 0.5. Next, we identified 738 immune-related genes that were expressed in the plasma of 479 patients with sepsis. Univariate Cox regression revealed that 222 genes were associated with prognosis (S6 Table),

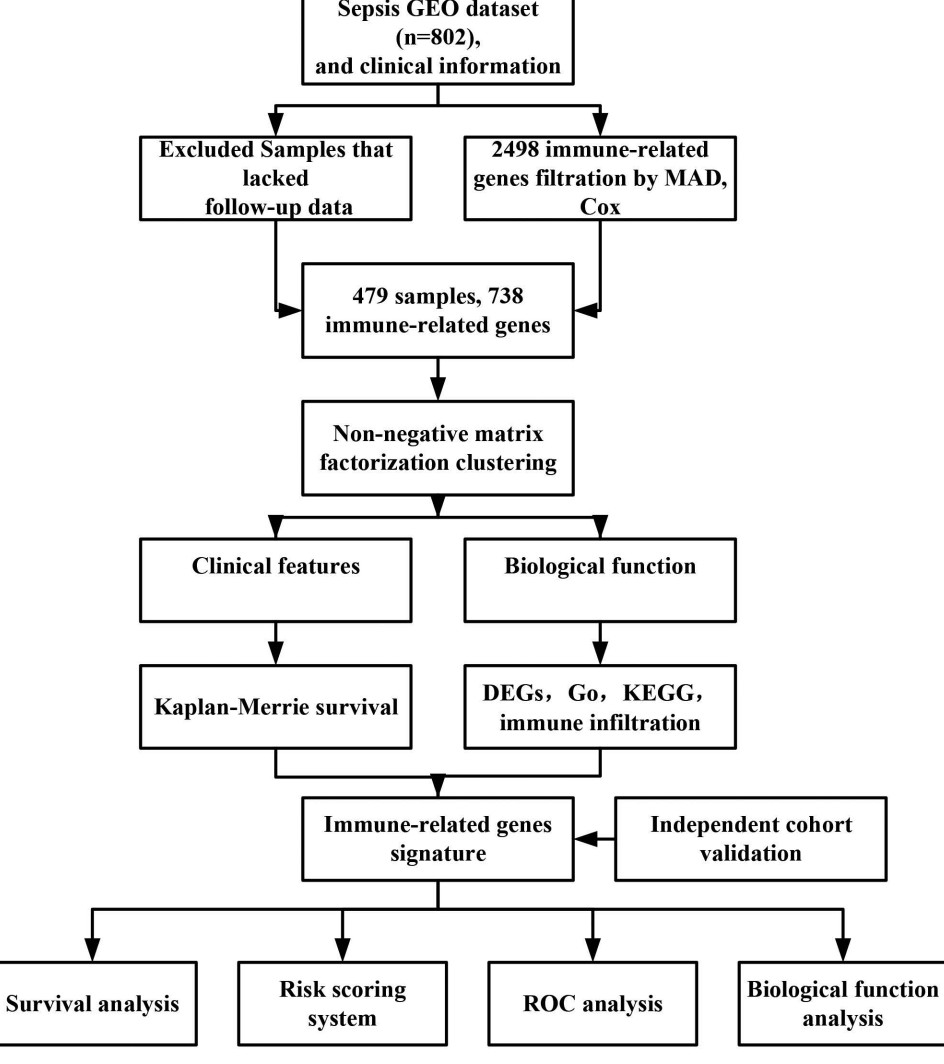

**Fig 1. The flow chart of the study.** Four hundred and seventy-nine samples with 738 immune-related genes were used for determining the molecular subtypes, establishing a prognosis model, and analyzing biological function in sepsis.

including 166 favorable genes and 56 risk genes. The top 50 genes are displayed in Fig 2A. The cophenetic correlation coefficient is the maximum, and cumulative distribution function is flatter when k=2. Importantly, the subtypes show significant differences in prognosis. When k>2, the consensus matrix heatmap shows unsharp and blur boundaries (S7 Fig). A k value of 2 was eventually chosen as the optimal number. This means that sepsis could be divided into two molecular subtypes (Fig 2B). The cumulative distribution function for k=2 was flatter and approached the maximum (Fig 2C). The PCA and t-SNE analysis further validated the classification of the two clusters (Fig 2D and 2E). The Kaplan–Meier analysis showed that patients in Cluster A had a poorer prognosis than those in Cluster B (Fig 2F, P<0.001).

## Biological characteristics of different molecular subtypes of sepsis

To explore the functional differences between the two molecular subtypes, we performed a GSVA. Cluster A was primarily enriched in pathways related to amino sugar and nucleotide sugar metabolism, peroxisomes, terpenoid backbone biosynthesis, p53 signaling, folate biosynthesis, glycine metabolism, autophagy regulation, sphingolipid metabolism, and O-glycan biosynthesis. In contrast, Cluster B was enriched in processes such as the Wnt signaling pathway, T cell receptor signaling, NK cell-mediated cytotoxicity, B cell receptor signaling, primary immunodeficiency, hematopoietic cell lineage development, and cell adhesion molecule synthesis (Fig 3A).

We also compared gene expression profiles between Cluster A and Cluster B, identifying 105 DEGs, with 55 upregulated and 50 downregulated genes (Fig 3B and 3C, S8 Table). GO enrichment analysis indicated that these DEGs were involved in processes such as cell killing, immune response regulation, T cell differentiation, and receptor binding (Fig 3D). The KEGG analysis revealed their involvement in immune-related pathways, including Th17, Th1, and Th2 cell differentiation; T cell receptor signaling; NK cell-mediated cytotoxicity; cytokine-cytokine receptor interaction; and the PD-L1/PD-1 checkpoint pathway (Fig 3E).

## Weighted gene co-expression network analysis of the molecular subtypes of sepsis

WGCNA identified the hub genes associated with the molecular subtypes of sepsis. The scale independence showed that $R^2$ reached its maximum and stabilized when the soft threshold was set to 6 (Fig 4A), with stable mean connectivity (Fig 4B). Dynamic tree cut identified 12 gene modules, with the gray module excluded from the classification (Fig 4C). Module-trait relationships showed that the Memagenta and MEtan modules were associated with Cluster B, while the MEgreen and MEsalmon modules were mainly linked to Cluster A (Fig 4D). A scatter plot demonstrated a high correlation (r=0.93) between gene significance for Cluster A and the magenta module (Fig 4E). We then constructed a protein-protein interaction (PPI) network using DEGs from Clusters A and B (Fig 5A). The hub genes identified via Cytoscape included IL2RB, CD3G, CD8A, CD3D, PRF1, CCR7, GZMB, GZMA, GAMK, and CCL5.

## Immune infiltration status of the different molecular subtypes

We assessed immune infiltration differences between Clusters A and B. The numbers of B cells, CD8+T cells, neutrophils, NK cells, pDCs, T helper cells, Tfh, Th1, Th2 cells, and TIL were significantly elevated in Cluster A, while the numbers of iDCs, macrophages, and Tregs were significantly lower in Cluster B compared to Cluster A (Fig 5B). Moreover, various immune functions, including APC co-stimulation, CCR signaling, checkpoint activity, cytolytic activity, HLA, inflammation-promoting responses, MHC class I, parainflammation, T cell co-inhibition and co-stimulation, and Type I IFN response, were elevated in Cluster B (Fig 5C). Comparing immune checkpoint gene expressions between Clusters A and B revealed that most immune checkpoint genes were significantly downregulated in Cluster A, except for CD200R1 and LAIR1, which were upregulated (Fig 5D).

## Establishment and validation of the immune-related gene prediction model for sepsis

We developed a prediction model using LASSO regression, identifying 36 potential genes from the univariate Cox regression analysis (Fig 6A and 6B). Multivariate Cox regression narrowed this down to 21 significant genes, which were

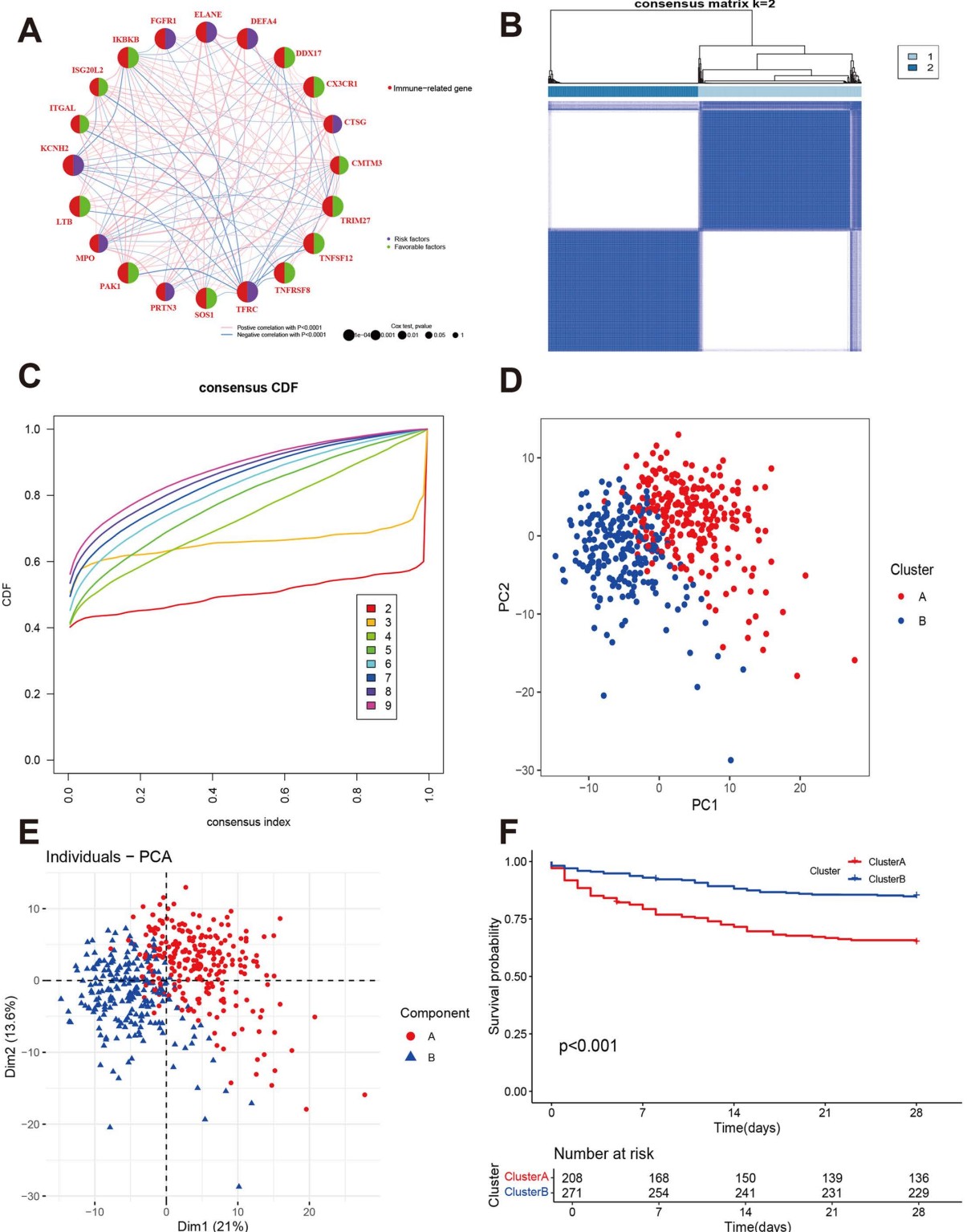

**Fig 2. Identification of the molecular subtypes of sepsis.** A: Network of the top 20 genes associated with prognosis. **B:** The consensus matrix was used to determine the optimal k value. **C:** The consensus cumulative distribution function plot. **D and E:** Principal component analysis and t-SNE of the molecular subtypes. **F:** Kaplan–Meier survival curves of the two molecular subtypes.

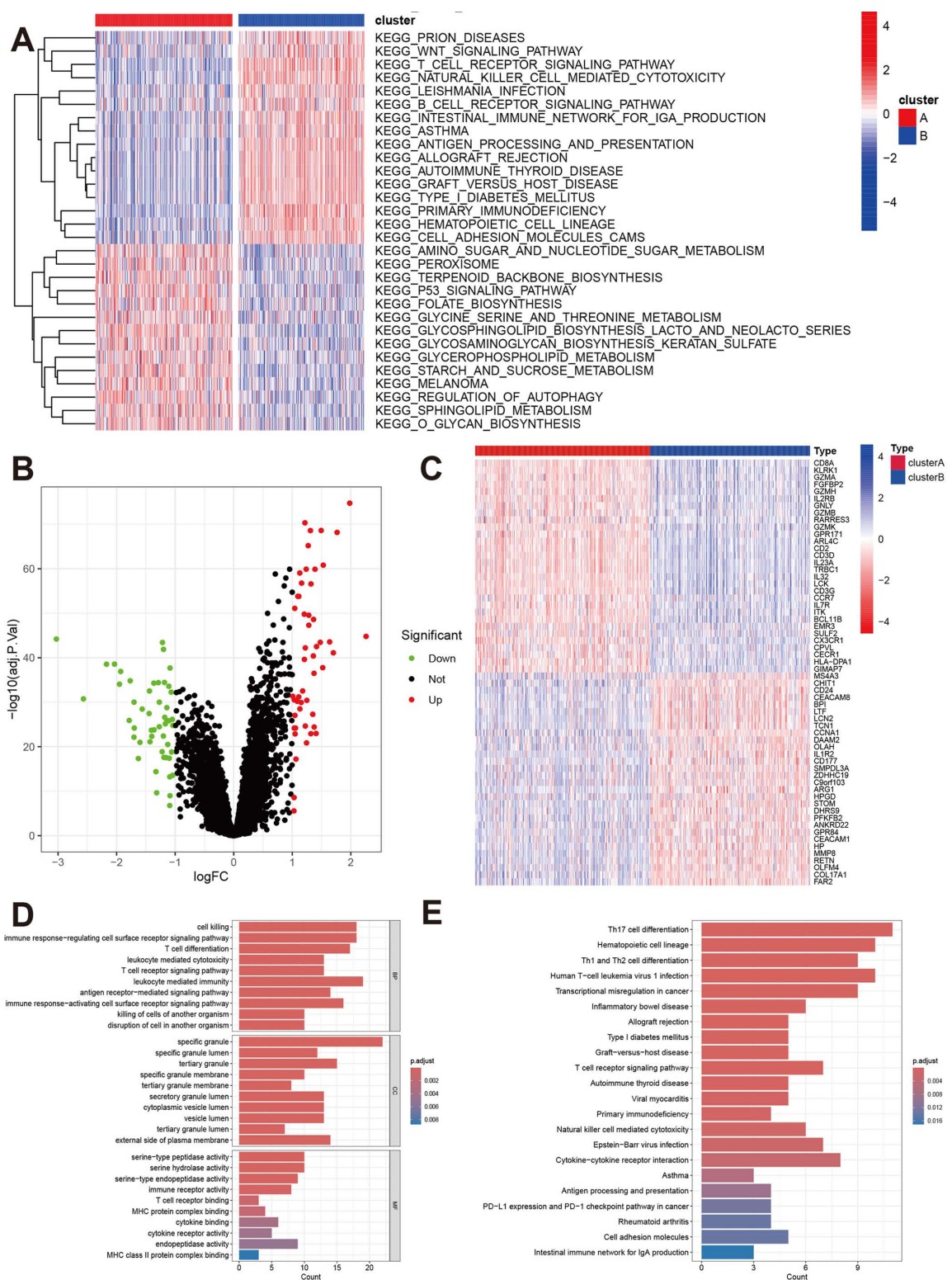

**Fig 3. Comparisons of biological function between the two molecular subtypes.** A: The GSVA shows the differences in signaling pathways. **B:** Volcano plot of the differently expressed genes between the two molecular subtypes. **C:** The heatmap shows the top DGEs. **D and E:** GO function and KEGG pathway analyses based on the DGEs between the two molecular subtypes.

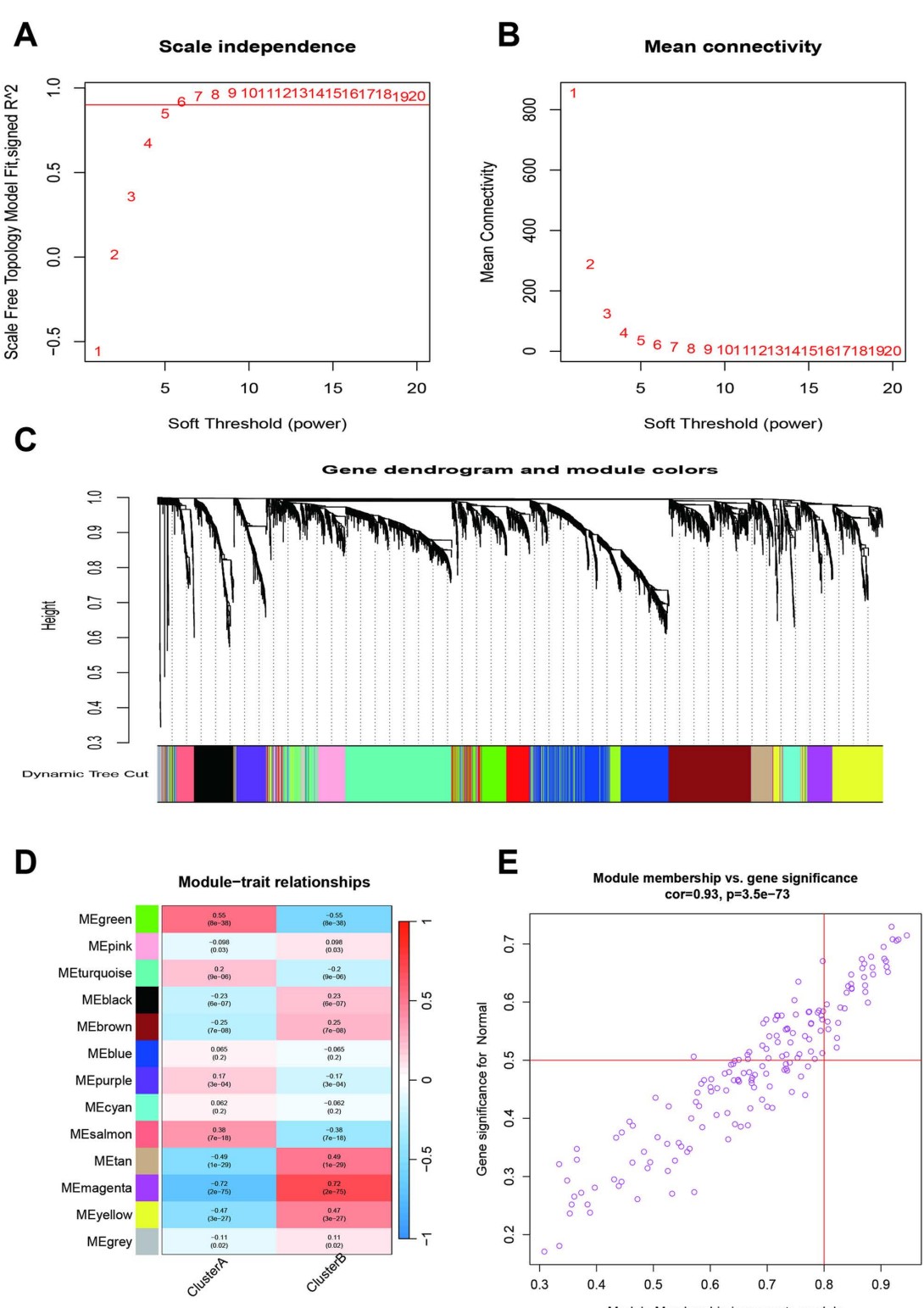

**Fig 4. Weighted gene co-expression network analysis for the molecular subtypes of sepsis. A and B:** The association between scale independence, mean connectivity, and soft threshold. **C:** Gene dendrogram and module colors. **D:** Module-trait relationships. **E:** Scatter plot of module membership in the magenta module.

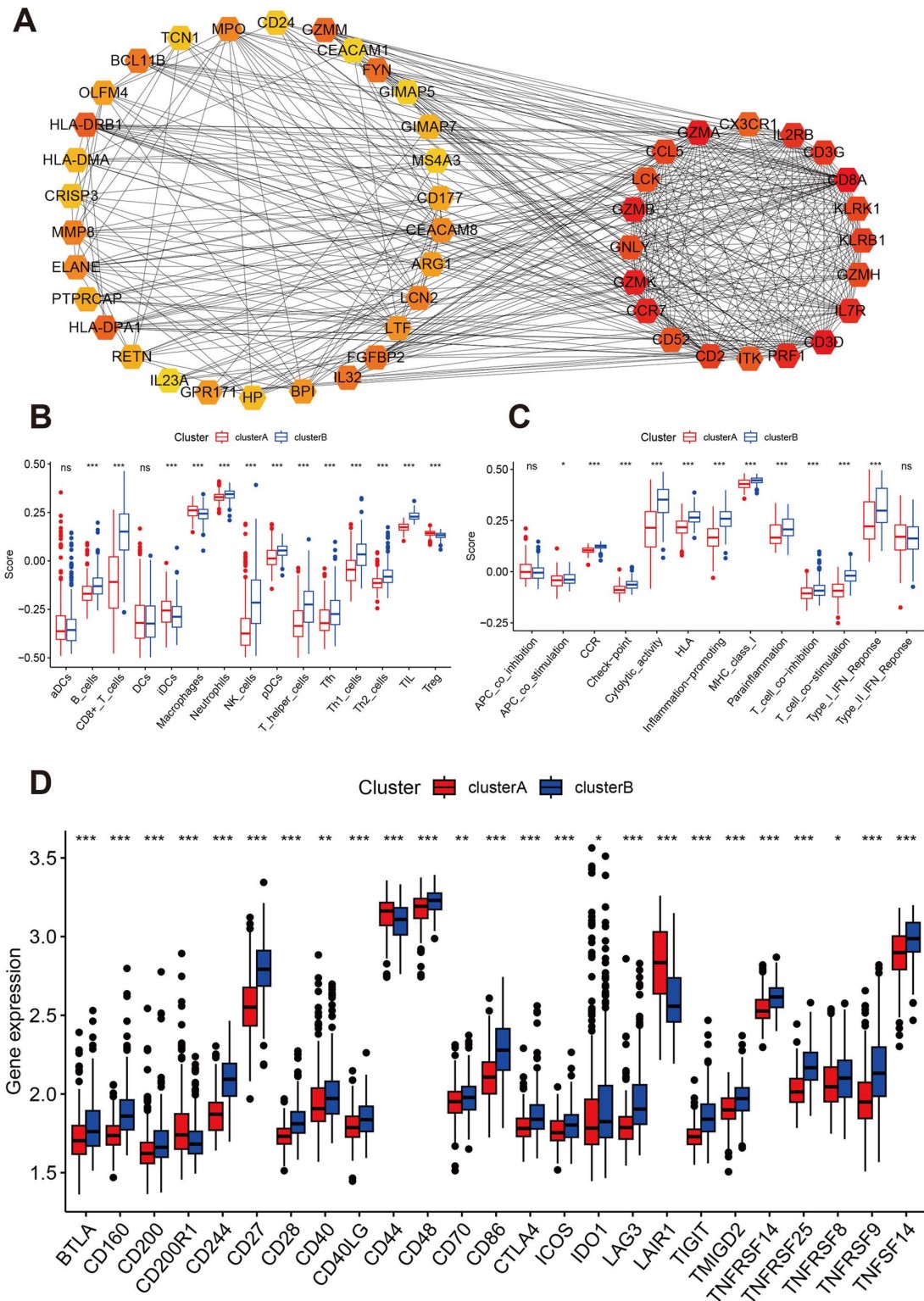

**Fig 5. Comparison of hub genes and immune cell infiltration levels between the two molecular subtypes. A:** Protein-protein interaction and the top 20 hub genes. **B and C:** Comparison of immune cell counts and function between Cluster A and Cluster B. **D:** Comparison of immune checkpoint gene expression levels between Cluster A and Cluster B.

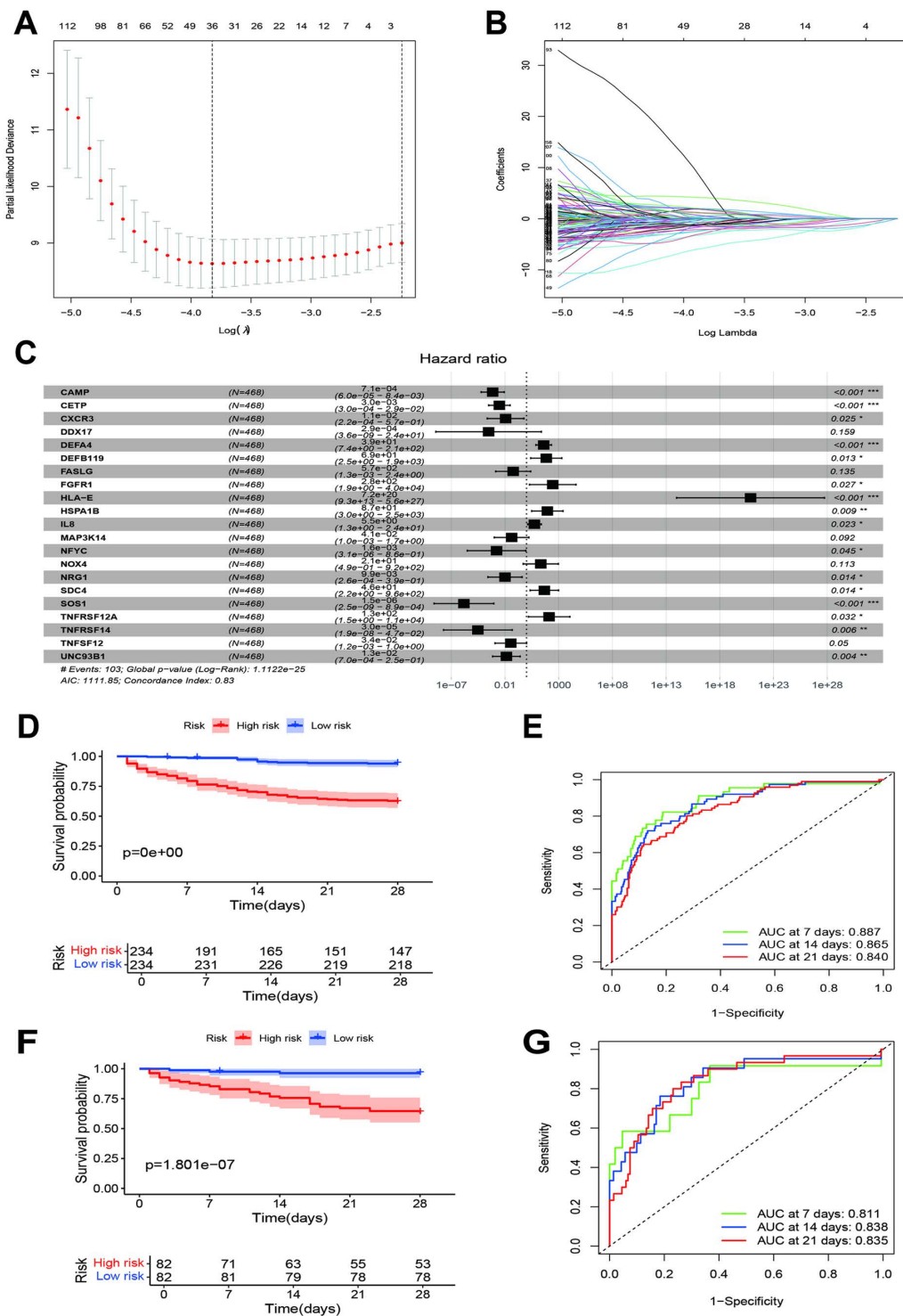

**Fig 6. Establishment and validation of an immune-related gene prediction model for sepsis. A and B:** The LASSO regression showed partial likelihood deviance and coefficients with different log lambda. **C:** Forest plot of the multivariate Cox regression analysis results. **D:** Kaplan–Meier survival curves of the high- and low-risk groups in training datasets. **E:** Receiver operating characteristics curves of the risk scores for the 7-day, 14-day, and 21-day outcomes in the training dataset. **F:** Kaplan–Meier survival curves of the high- and low-risk groups in the validation dataset. **G:** Receiver operating characteristics curves of the risk score for the 7-day, 14-day, and 21-day outcomes in the validation dataset.

included in the final model (S9 Table), with a concordance index of 0.83 (Fig 6C). We calculated a risk score for each patient based on the regression coefficients and gene expression levels. Patients in the high-risk group had significantly poorer prognoses than those in the low-risk group (Fig 6D). The AUCs of the risk score predicting the 7-day, 14-day, and 21-day outcomes were 0.887, 0.865, and 0.840, respectively, in the training dataset (Fig 6E). This model was validated in an independent cohort, yielding similar results (Fig 6F). The AUCs of the risk score predicting the 7-day, 14-day, and 21-day outcomes were 0.811, 0.838, and 0.835, respectively, in the validation dataset (Fig 6G). Stratified analysis across clinical parameters further confirmed that patients in the high-risk group had worse prognoses (Fig 7A–7Q).

## Independent prognosis analysis and risk scoring system

The comparisons of clinical characteristics between training dataset and validation dataset were presented in Table 1. There are no significant differences in mortality, age, and gender. The significant differences were observed in the ratios of pneumonia, thrombocytopenia, infection type, and diabetes.

Univariate Cox regression revealed that the risk score was associated with poor prognosis in both the training (HR: 1.043, 95% CI: 1.035–1.050, $P < 0.001$; Fig 8A) and validation (HR: 1.041, 95% CI: 1.028–1.055, $P < 0.001$; Fig 8B) datasets. Multivariate Cox regression confirmed that the risk score was an independent predictor in both the training (HR: 1.0466, 95% CI: 1.037–1.055, $P < 0.001$; Fig 8C) and validation (HR: 1.038, 95% CI: 1.022–1.053, $P < 0.001$; Fig 8D) datasets. The calculated power is 92.3% based on risk score, suggesting the sample size sufficiency.

A risk scoring system was developed based on the risk score and clinical parameters, with estimated mortality probabilities of 0.864 (7-day), 0.784 (14-day), and 0.734 (21-day) (Fig 8E). Model evaluation demonstrated an AUC of 0.865 for the risk score at 14 days (Fig 8F), and calibration plots showed good model fit across 7, 14, and 21 days (Fig 8G).

## Biological function and immune status of high- and low-risk groups

We identified 353 DEGs between the high- and low-risk groups (S10 Table). The GO enrichment analysis suggested that these genes were primarily involved in immune responses (Fig 9A). The KEGG analysis highlighted immune-related pathways, including Th1, Th2, and Th17 cell differentiation; cell adhesion molecule function; NOD-like receptor signaling; and T cell receptor signaling pathways (Fig 9B). The immune status analysis showed that the B cell, CD8 + T cell, neutrophil, NK cell, pDC, T helper cell, Th1 cell, Th2 cell, and TIL counts were elevated in the low-risk group, while DCs, iDCs, and Tregs were more prominent in the high-risk group (Fig 9C). The immune function analysis revealed higher levels of CCR signaling, checkpoint activity, cytolytic activity, HLA, inflammation-promoting responses, MHC class I, cell co-inhibition and co-stimulation, and type II IFN responses in the high-risk group (Fig 9D). Most immune checkpoint genes were highly expressed in the low-risk group, except for CD276, ICOSLG, LAIR1, and PDCD1LG2, which were elevated in the high-risk group (Fig 9E). The correlation analysis indicated that eosinophils; M0, M1, and M2 macrophages; resting mast cells; resting NK cells; plasma cells; and CD4 naïve T cells were positively correlated with the risk score (Fig 10A–10H), whereas activated dendritic cells, activated NK cells, monocytes, and CD8+ T cells showed positive correlations (Fig 10I–10L).

## Discussion

The main findings of this study are as follows: (1) This study identified two molecular subtypes of sepsis based on gene expression profiles. Cluster A showed poorer prognosis than Cluster B, and further analyses revealed differences in biological pathways and immune characteristics between these subtypes. (2) Cluster A was enriched in metabolic pathways (amino sugar metabolism and p53 signaling), while Cluster B was associated with immune-related pathways (Wnt signaling and T and B cell receptor signaling). One hundred and five DEGs were identified between the two clusters, with Cluster A having higher expression of immune response-regulating genes. (3) Immune infiltration analysis showed that Cluster B had higher immune cell counts (CD8+ T cells, NK cells) than Cluster A, while immune-suppressing cell counts

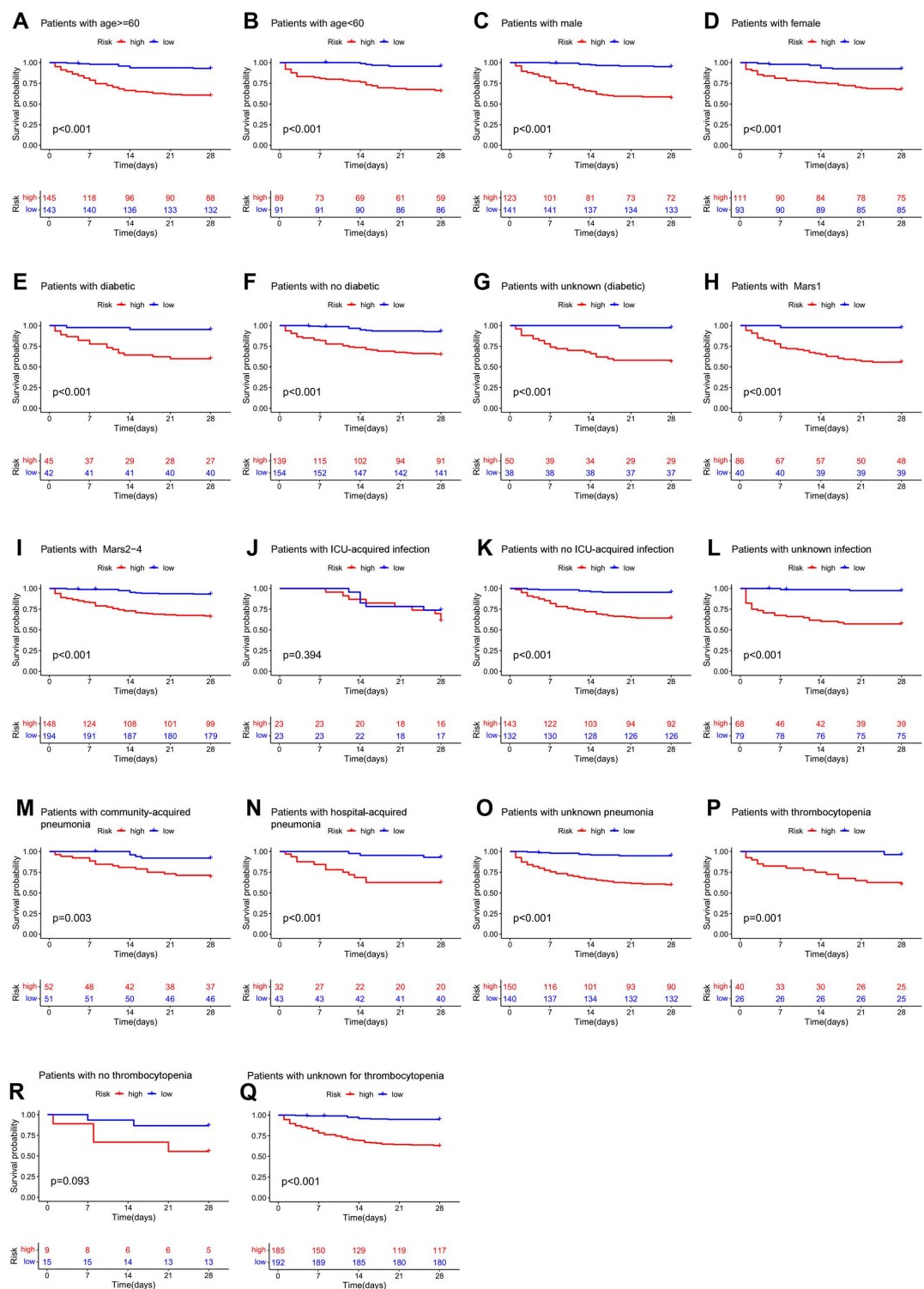

**Fig 7. Kaplan–Meier survival curves of the stratified analysis of the high- and low-risk groups. A and B:** Age (< 60 and ≥ 60). **C and D:** Male and female. **E, F and G:** No diabetic, diabetic and unknown. **H and I:** Mars1 and Mars (2-4). **J, K and L:** No ICU-acquired infection, ICU-acquired infection and unknown infection. **M, N and O:** Community-acquired pneumonia, hospital-acquired pneumonia and unknown pneumonia. **P, R and Q:** Thrombocytopenia, no thrombocytopenia and unknown for thrombocytopenia.

**Table 1. General characteristics of training and validation datasets.**

| Variable | Training group | Validation group | *P* |
|---|---|---|---|
| Mortality (N, %) | 103(22.0%) | 32(19.8%) | 0.547 |
| Gender (Male, %) | 264(56.4%) | 94(58.0%) | 0.721 |
| Age (>60 years, %) | 288(61.5%) | 95(58.6%) | 0.515 |
| Pneumonia | | | 0.024 |
| Community-acquired | 103(22.0%) | 74(45.7%) | |
| Hospital-acquired | 75(16.0%) | 88(54.3%) | |
| Unknown | 290(62.0%) | 0(0.0%) | |
| Thrombocytopenia | | | <0.001 |
| Yes | 66(14.1%) | 74(45.7%) | |
| No | 25(5.3%) | 88(54.3%) | |
| Unknown | 377(80.6%) | 0(0.0%) | |
| Infection type | | | <0.001 |
| Yes | 46(9.8%) | 51(31.5%) | |
| No | 275(58.8% | 111(68.5%) | |
| Unknown | 147(31.4%) | 0(0.0%) | |
| Endotype | | | – |
| Mars1 | 126(26.9%) | – | |
| Mars2–4 | 342(73.1%) | – | |
| Diabetes | | | <0.001 |
| Yes | 87(18.6%) | 83(51.2%) | |
| No | 293(62.6%) | 79(48.8%) | |
| Unknow | 88(18.8%) | 0(0.0%) | |
| APACHEII | – | 19.88±3.4 | – |

were lower in Cluster B than in Cluster A. (4) An immune-related gene prediction model was developed and validated, showing that patients with a higher risk score had worse prognosis. The risk scoring system, based on gene expression and clinical data, showed high accuracy in predicting mortality. (5) High-risk patients had elevated immune cell levels and immune functions, while low-risk patients had higher expression of certain immune checkpoint genes. Overall, this study provides insights into the molecular subtypes of sepsis, offering the potential for personalized prognosis assessment and treatment strategies based on immune-related pathways and gene expression profiles.

Many studies have been conducted on predictive models for assessing the prognosis of sepsis, including pyroptosis-related genes [28], endoplasmic reticulum stress genes [29], hypoxia- and lactate metabolism-related genes [30], inflammatory response-related genes [31], and platelet-related genes [32]. The predictive abilities of these models varied widely. A recent study also analyzed the associations between immune-related genes and clinical prognosis or the immune microenvironment in sepsis [18]. There are evident differences between our research and previous studies that we need to address. First, although we utilized the same dataset, our research diverges from previous studies in terms of data processing strategies. While prior studies divided the GEO dataset into training and validation sets for model construction, our study used the GEO dataset exclusively as a training set and incorporated an independently collected validation cohort. Secondly, our research performed an unsupervised clustering analysis on the GEO dataset, identifying two molecular subtypes with significantly distinct clinical prognoses, biological functions, and immunologic backgrounds, which had not been done in earlier studies. Furthermore, our model differs from those developed in prior research; whereas previous studies created a prognostic model comprising seven immune-related genes with a consistency index of 0.78, our model includes twenty-one immune-related genes and achieves a consistency index of 0.83. In terms of predictive ability, our

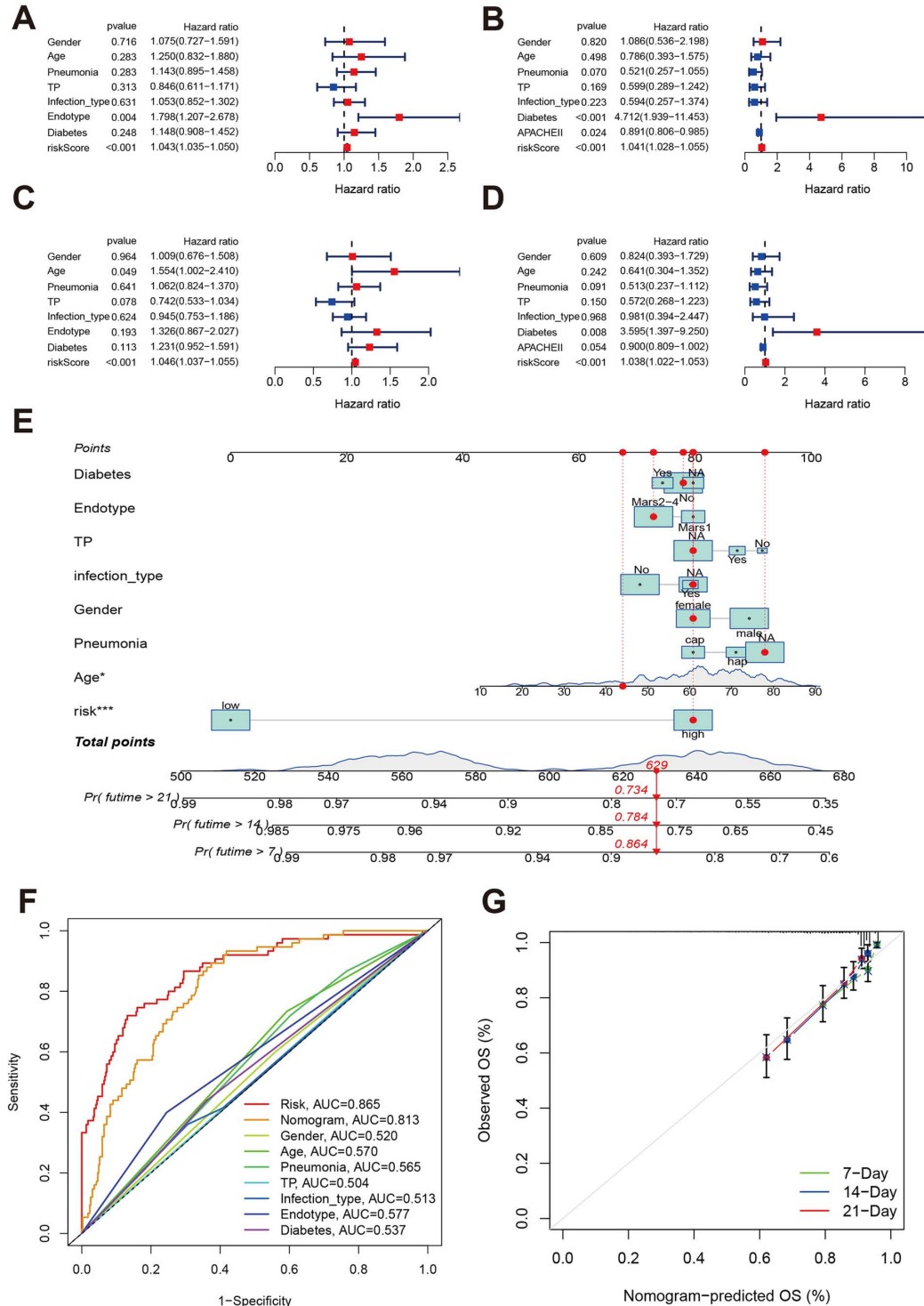

**Fig 8. Independent prognosis analysis and the risk scoring system. A and B:** Univariate Cox regression analysis of the risk score and prognosis of patients with sepsis in the training and validation datasets. **C and D:** Multivariate Cox regression analysis of the risk score and prognosis of patients with sepsis in the training and validation datasets. **E:** Nomogram plot estimating an individual's mortality risk. **F:** Time-independent receiver operating characteristics curves of the risk score and clinical parameters. **G:** The calibration plot shows the degree of fit of the observed OS and predicted OS at 7, 14, and 21 days.

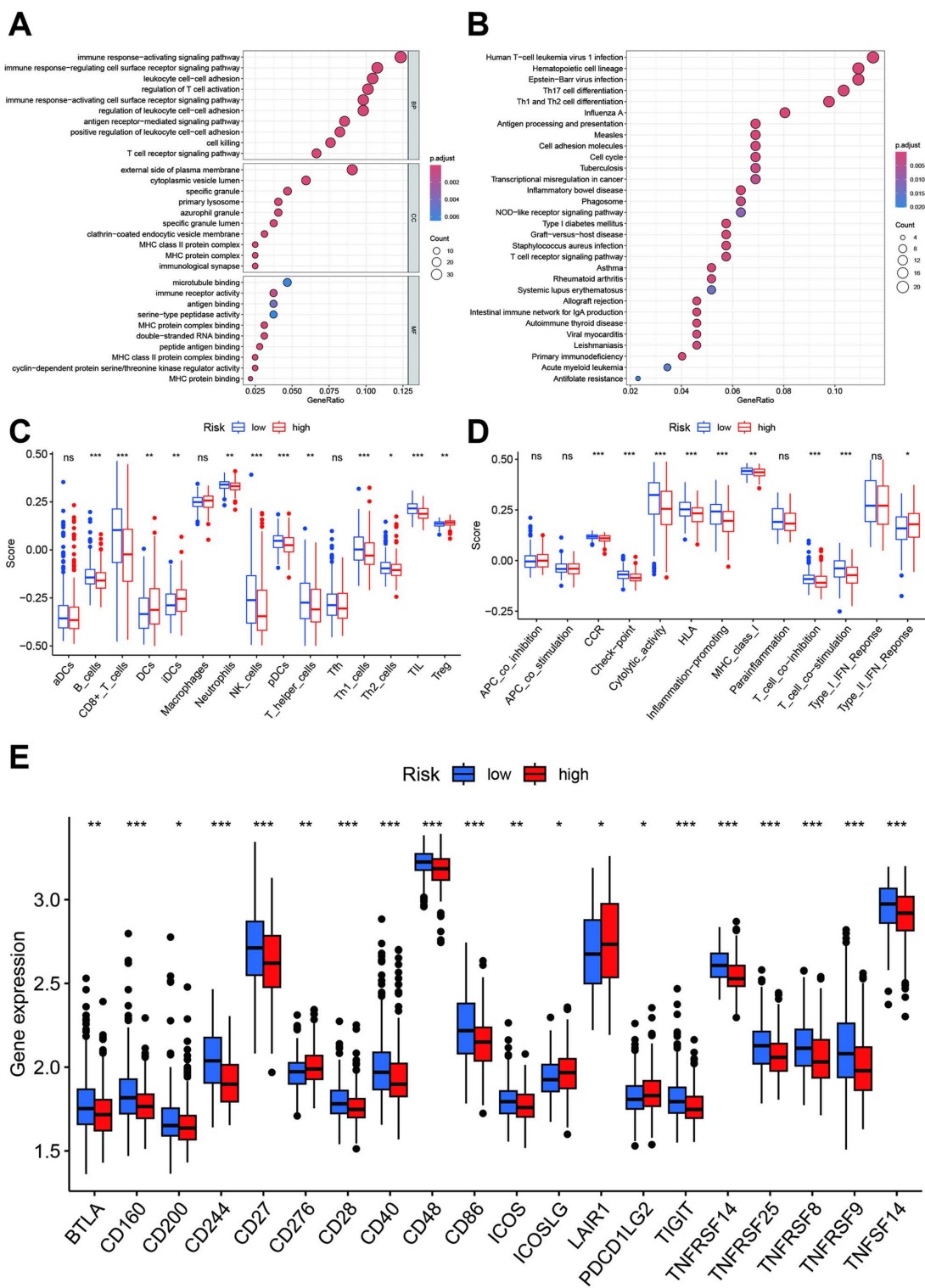

**Fig 9. Biological function and immune status of the high- and low-risk groups. A and B:** GO function and KEGG pathway analyses based on the DGEs between high- and low-risk groups. **C and D:** Differences in immune cells and function between the high- and low-risk groups. **E:** Differences in the expression levels of immune checkpoint genes between the high- and low-risk groups.

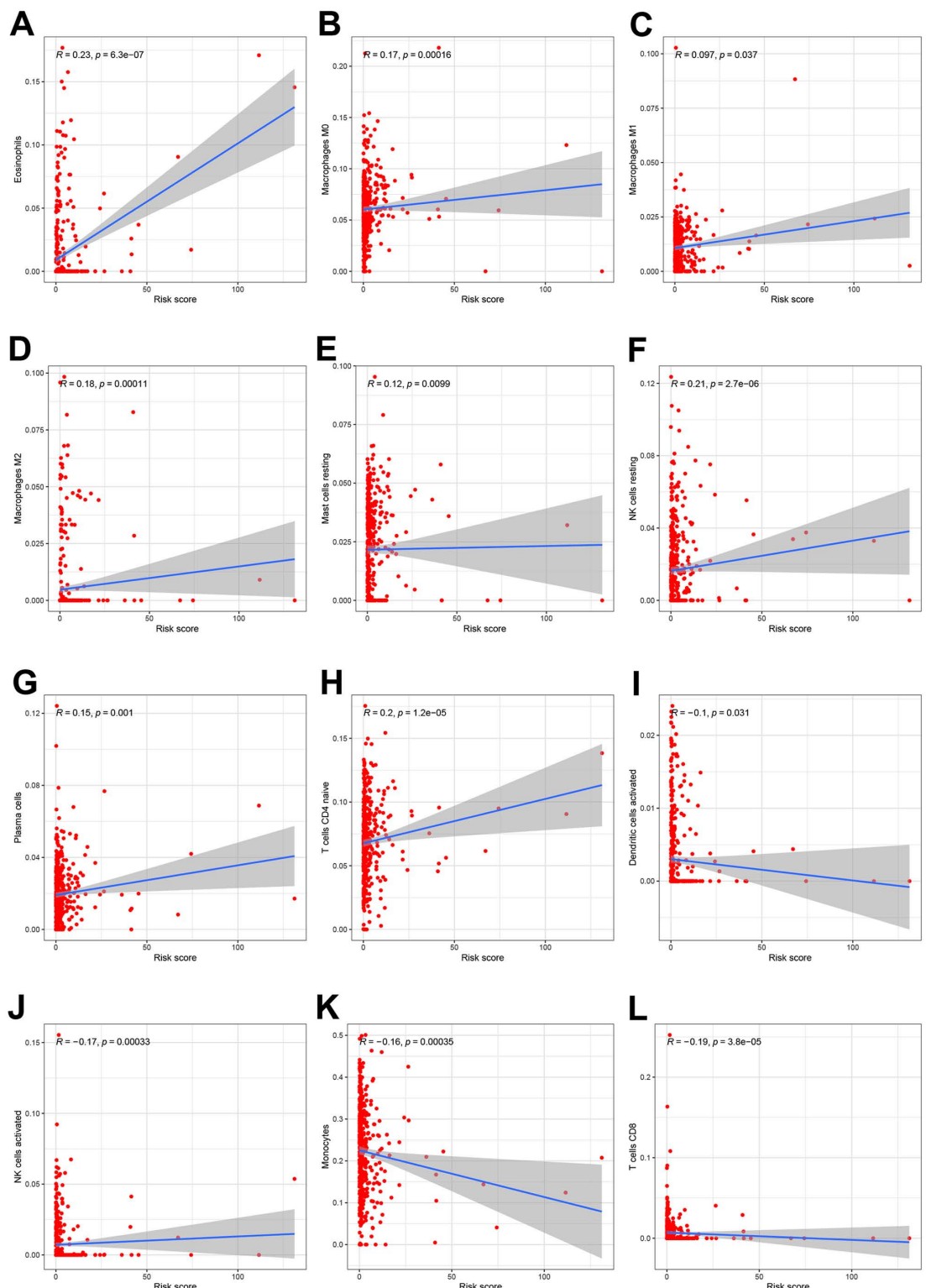

**Fig 10. Association between the risk score and immune cells. A: Eosinophils. B: M0 macrophages. C:** M1 macrophages. **D:** M2 macrophages. **E:** Resting mast cells. **F:** Resting NK cells. **G:** Plasma cells. **H:** CD4 naïve T cells. **I:** Activated dendritic cells. **J:** Activated NK cells. **K:** Monocytes. **L:** CD8+ T cells.

model's predictive ability was relatively high. Finally, our research established a personalized risk scoring system based on risk scores and estimated the mortality risk for patients at each critical time point. Additionally, we connected the molecular subtypes of patients with sepsis to the prognostic risk patterns, forming a "molecular subtype-risk identification-risk assessment" system. Some other gene prognosis models were also developed. Yang et al developed a nine-gene model and the AUCs were 0.83, 0.81, and 0.77 at 7, 14 and 21 days. But the internal and external validation datasets were lacking for this study, and the molecular subtypes were also not identified [33]. Peng developed a hypoxia- and lactate metabolism-related gene signature to predict prognosis of sepsis, which included five genes. Although this study includes an external validation dataset, the predictive ability of the model is moderate, and the AUC is 0.674. Similarly, this study did not identify the molecular subtypes [30]. Liang et al built a six-gene based on pyroptosis genes prognostic model with an external validation dataset. However, this predictive ability of this mode is also low, and the AUCs were 0.66, 0.63, and 0.64 at 7 days, 14 days and 28 days [28]. Compared to these studies, our model has better predictive power and a richer presentation of results. Our study not only enhances but also expands upon prior research, providing robust supporting data for validation.

It has been suggested that sepsis exhibits significant heterogeneity, necessitating its classification into subtypes to better understand its clinical characteristics and prognosis [18]. More importantly, this classification aims to enable precise treatment for different subtypes, thereby improving the outcomes of sepsis. Our research identified two molecular subtypes of sepsis based on prognosis-related immune genes: cluster A and cluster B. Cluster A had a significantly worse prognosis than cluster B, and the two subtypes showed substantial differences in immune characteristics, biological functions, and immune cell infiltration levels. Specifically, almost all immune cells and immune function were significantly elevated in Cluster B except the following immune cells (iDCs, macrophages, and Tregs). In sepsis, both the function and the number of various immune cells undergo significant changes, leading to an immunosuppressive state. These alterations include shifts in cell counts, functional impairments, and imbalances in the secretion of pro-inflammatory and anti-inflammatory factors. Together, these disruptions contribute to immune system dysfunction, which adversely impacts the clinical outcomes of patients. This study found that patients in cluster A had a worse prognosis compared to those in cluster B. Immune infiltration analysis revealed that most immune cells and their functions were suppressed in cluster A patients, and the expression of immune checkpoint inhibitory genes was relatively elevated, supporting the notion of immune system suppression in sepsis [34]. Similar trends were observed in high-risk and low-risk groups. However, some immune cells and functions, such as iDCs, macrophages, and Tregs, were elevated in patients with a poor prognosis, particularly in cluster A. This may be linked to their specific roles: iDCs, although they have a strong capacity for antigen uptake, are functionally inactive and have a limited ability to activate T-cell responses [35]; macrophages primarily regulate non-specific immunity and help maintain immune tolerance. Their increased presence may represent a stress response [36]. Immune checkpoints are molecules expressed on the surface of immune cells that regulate the degree of immune activation, acting as a "brake" to keep immune responses within normal limits [37]. The abnormal expression and function of immune checkpoints are important contributors to the development of various diseases. In this study, we observed increased expression of several immune checkpoint genes, CD200R1 and LAIR1, both of which are immunosuppressive. This upregulation is associated with the immunosuppressive effects seen in sepsis patients. In the high-risk group, LAIR1 was also elevated, as were immunosuppressive genes CD276, ICOSLG, and PDCD1LG2, which is consistent with immunosuppressive status in sepsis patients and further indicates that poor prognosis in patients with sepsis may be closely linked to immunosuppression.

Currently, there is no standardized set of criteria or methodology for the study of sepsis subtypes. Previous research has also classified sepsis based on genomic markers. In Davenport et al.'s study on sepsis caused by community-acquired pneumonia, two subtypes were identified (SRS1 and SRS2), with SRS1 exhibiting more severe immunosuppression and higher mortality rates than SRS2 [38]. In a study involving 306 patients with sepsis, four subtypes were established, with Mars 1 having the most severity and highest mortality rate; this subtype was associated with specific

genes (BPGM and TAP2) [39]. Wong et al. classified pediatric sepsis into two subtypes (A and B), with type A being more severe and significantly benefitting from corticosteroid treatment [40]. In our study, we also obtained two subtypes (cluster A and cluster B) based on immune-related genes, and the cluster A had poorer prognosis than cluster B. Our study provided a different perspective for sepsis subtypes. In multivariate analyses, we found confirmed that the risk score based on immune-related genes was an independent prognosis factor in both training and validation datasets. We also found that age was independently associated with poor prognosis in sepsis. Older adults are immunocompromised, making them more vulnerable to sepsis, and their prognosis tends to be poorer. We did not find the correlation of gender with prognosis in sepsis patients. Previous study suggested that gender may influence age-associated sepsis outcomes. We assume that there mab be interactions between age and gender, which need to be further confirmed [41]. We also found no association between endotype and prognosis in sepsis patients. Endotype is genotyped based on the genome, but sepsis is characterized by rapid disease progression, requiring quick clinical decision-making. Genotype-based endotyping may therefore have certain time lags. Additionally, the variation in patient characteristics could play a role. For instance, sepsis patients often present with immunosuppression, rapid disease progression, and the administration of specific treatments, all of which may influence the relationship between molecular typing and prognosis. Sepsis is a syndrome caused by different infectious agents, and it results from life-threatening organ dysfunction. Therefore, a "one-size-fits-all" approach to its treatment is inappropriate, and it may require precise treatment based on the subtype. Research into sepsis subtypes has paved the way for precision medicine in sepsis treatment, and the heterogeneity in treatment response among different subtypes gives hope for more targeted sepsis therapies.

In sepsis, aside from the early inflammatory response to infection or, in some cases, an excessive inflammatory reaction, severe immunosuppression caused by a compensatory anti-inflammatory response is a leading cause of sepsis-related mortality [42]. The host's immune dysregulation is a key mechanism driving the onset and progression of sepsis. Our study found that, compared to the low-risk group, the high-risk group exhibited enrichment in numerous immune-related signaling pathways and functions, such as the NOD-like receptor signaling, immune response-activating signaling, immune response-regulating cell surface receptor signaling, immune response-activating cell surface receptor signaling, and T cell receptor signaling pathways. NOD-like receptor signaling is closely associated with the formation of inflammasomes, which, through the classical pathway, promote the maturation and secretion of IL-1β and IL-18 via activated caspase-1. Extracellular IL-1β enhances the pro-inflammatory effects of IL-6 and TNF-α, while IL-18 mediates neutrophil maturation and local infiltration and induces adaptive immunity to shift toward the Th2 response. Th2 cells secrete IL-4 and IL-13, which drive macrophages toward the M2 phenotype, promoting an anti-inflammatory effect [43–45]. The T cell receptor signaling pathway is finely regulated by positive and negative regulatory mechanisms, and its balance is crucial to the immune response. In negative regulation, inhibitory receptors such as CTLA4 and PD-1 play a particularly significant role and have become key targets of anti-tumor therapy, which could be useful in sepsis treatment [46]. It is generally believed that sepsis is an organ dysfunction caused by a dysregulated immune response to infection. The immune response-activating cell surface receptor signaling pathways were closely related to immune signaling transmission, which further suggests that immunity plays an important role in sepsis. During acute inflammation, inflammasome activation helps clear damaged cells and initiate tissue repair, whereas prolonged activation in chronic inflammation can harm host tissues. These two types of inflammation can be modulated by regulating inflammasome activity [47]. In this study, we observed that the levels of Th1 and Th2 cell were decreased in high-risk group. Additionally, during sepsis-induced immunosuppression, lymphocyte depletion and exhaustion occur, along with the expansion of anti-inflammatory immune cells, decreased expression of human leukocyte antigen DR (HLA-DR), increased expression of programmed death-1, metabolic alterations, and epigenetic alterations. These mechanisms collectively contribute to a complex immunosuppressive environment, increasing the risk of secondary infections and death [48]. We found that the B cell, CD8+T cell, neutrophil, NK cell, pDC, T helper cell, Th1 cell, Th2 cell, and TIL counts were elevated in the low-risk group but decreased in the high-risk group, suggesting that the immune status of high-risk patients was suppressed. This may be related to T cell exhaustion. A previous study investigated the leukocyte functions

before and after passive blocking of PD-1 and PD-L1 with antibodies. White blood cells were derived from patients with sepsis, critically ill patients without sepsis and healthy controls. They found that neutrophil and monocyte functions in patients with sepsis are decreased, which is related to CD8＋T cell- and NK cell-specific PD-1 and PD-L1 expression [49]. Interestingly, decreased CD8＋T cell effector function (such as IFNγ production capacity) was associated with elevated PD-L1 expression in neutrophils. After the application of anti-PD-1 or PD-L1 antibodies, the function of neutrophils and monocytes was restored, leukocyte apoptosis was reduced in patients with sepsis, and the function of T cell effects was also significantly restored, manifested by a significant increase in IFN-γ and IL-2 production [50]. This provides evidence for T cell exhaustion in patients with sepsis, but it is reversible.

This study, while providing valuable insights into immune-related molecular subtypes and a predictive model for sepsis, has several limitations. First, the training dataset was derived from a single public database (GEO), which may introduce bias based on sample selection and population differences. Although an independent cohort was used for validation, both datasets may not fully capture the heterogeneity of sepsis across different patient populations, limiting the study's generalizability. To minimize this, we ensured that the clinical information in the validation set matched that of the training set. We made efforts to keep both sets homogeneous in clinical characteristics, including the sex ratio, age distribution, pneumonia type, and infection type, as demonstrated by the multivariate Cox regression analysis. These clinical variables were not associated with prognosis in either dataset. Additionally, the two datasets represent individuals from different ethnic groups, which may improve the model's applicability to some extent. Second, the variables (pneumonia, thrombocytopenia, infection type, and diabetes) had some missing data. We included these variables in the multivariate cox regression to make full use of this dataset. This may generate some information bias. The stratified analyses in the missing data and results from the validation dataset weaken the effects of missing data on results. This study did not also collect information regarding the specific organisms responsible for sepsis. As a result, we were unable to assess the impact of different sepsis-causing pathogens on the pathogenesis and progression to severe sepsis. This is an important factor that could influence outcomes, and we recommend that future studies with more comprehensive datasets consider incorporating pathogen-specific data to better understand these dynamics. Third, while the immune-related genes used to construct the prognostic model were identified through bioinformatics approaches, experimental validation beyond qPCR is necessary to confirm their biological roles in sepsis. Furthermore, the study focused solely on immune-related genes, potentially overlooking other pathways that contribute to sepsis progression. Lastly, while the risk scoring system shows promise, external validation with larger, multicenter cohorts is required before it can be applied in clinical settings.

This study identified distinct immune-related molecular subtypes of sepsis and developed an immune-related gene prognostic model that divides patients into high- and low-risk groups based on immune response patterns. These findings enhance our understanding of immune dysregulation in sepsis and provide a foundation for personalized treatment strategies. The proposed risk scoring system can potentially improve sepsis prognosis and guide clinical decision-making. Different intervention measures should be taken for different subtypes and risk populations. However, further research is needed to validate these findings in broader patient populations and to understand potential molecular mechanisms of sepsis through systematic wet-lab validation.

## Supporting information

**S1 File. The tripod checklist.**
(DOCX)

**S2 File. Original data.**
(XLSX)

**S3 File. Original code.**
(TXT)

**S4 Table. The lists of immune-related genes.**
(XLS)

**S5 Table. The primer design of 21 genes for qPCR.**
(XLSX)

**S6 Table. Univariate cox regression identified the prognosis-related immune genes.**
(XLSX)

**S7 Fig. The matrix heatmaps when K > 2.**
(TIF)

**S8 Table. Differentially expressed genes between cluster A and cluster B.**
(XLS)

**S9 Table. Multivariate cox regression for prediction model.**
(XLSX)

**S10 Table. Differentially expressed genes between high-risk and low-risk groups.**
(XLS)

## Author contributions

**Conceptualization:** Changmin Wang.

**Data curation:** Zhiwei Li, Shuting Yang, Yezi Liu.

**Formal analysis:** Leyi Wang, Shuting Yang.

**Funding acquisition:** Changmin Wang.

**Investigation:** Mengsi Chen.

**Methodology:** Bin Luo, Yezi Liu, Mengsi Chen.

**Project administration:** Changmin Wang.

**Software:** Bin Luo, Mengsi Chen.

**Validation:** Zhiwei Li, Bin Luo.

**Visualization:** Leyi Wang, Yezi Liu.

**Writing – original draft:** Zhiwei Li.

**Writing – review & editing:** Leyi Wang, Changmin Wang.

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
