## [Decision Letter · Decision Letter 0]

PONE-D-24-57866Immune-associated molecular classification and prognosis signature of sepsis

PLOS ONE

Dear Dr. Wang,

Thank you for submitting your manuscript to PLOS ONE. After careful consideration, we feel that it has merit but does not fully meet PLOS ONE’s publication criteria as it currently stands. Therefore, we invite you to submit a revised version of the manuscript that addresses the points raised during the review process.

We look forward to receiving your revised manuscript.

Kind regards,

Nishel Mohan Shah, PhD

Academic Editor

PLOS ONE

3. For studies involving third-party data, we encourage authors to share any data specific to their analyses that they can legally distribute. PLOS recognizes, however, that authors may be using third-party data they do not have the rights to share. When third-party data cannot be publicly shared, authors must provide all information necessary for interested researchers to apply to gain access to the data. (https://journals.plos.org/plosone/s/data-availability#loc-acceptable-data-access-restrictions)

Additional Editor Comments:

Please note that I have acted as a reviewer for this manuscript, and you will find my comments below, under Reviewer 2.

Reviewers' comments:

Reviewer's Responses to Questions

**Comments to the Author**

1. Is the manuscript technically sound, and do the data support the conclusions?

Reviewer #1: Yes

Reviewer #2: Yes

2. Has the statistical analysis been performed appropriately and rigorously? 

Reviewer #1: Yes

Reviewer #2: Yes

3. Have the authors made all data underlying the findings in their manuscript fully available?

Reviewer #1: Yes

Reviewer #2: Yes

4. Is the manuscript presented in an intelligible fashion and written in standard English?

Reviewer #1: Yes

Reviewer #2: Yes

5. Review Comments to the Author

Reviewer #1: This is a very complex study involving a considerable amount of work with clinically important objectives.

The manuscript is well written and the discussion is argued convincingly.The discussion of the pros and the limitations of the study were insightful. Could the authors consider adding further specific detail about the findings of other similar prior studies in the discussion?

I have some minor questions:

1. Could the authors expand in the discussion regarding any paradoxes that there may be in the biological function and immune status (immune cells and checkpoint genes) seen between cluster A (Figures 5B, C&D) and the high risk groups (Fig 9C, D & E).

2. Could the authors expand in the discussion their hypotheses regarding the poor prognostic performance of variables that have been associated with poor prognosis in prior phenotypic studies e.g., age, sex.

3. Also, it may be interesting to expand on the poor performance of "endotype" in both the training and validation datasets.

I have some minor suggestions re: visualisation of the data:

4. Figure 2A - the labelling is not clearly visualised for a number of the genes in this figure, could the resolution of this figure be optimised? Could this be achieved by reducing the number of genes in the figure to only those with a cox p value of 0.05 or less? Could changing the colour of the negative correlations from pink also improve the resolution of correlations.

5. Maintaining consistency of the colours representing clusters A and B in all figures and graphs could be considered to help the reader easily identify the relevant cluster.

6. Is the unit for K-M (Fig 2F) intended to be days rather than years?

Reviewer #2: Review

Overall, an informative and interesting study. The study provides valuable insights into sepsis prognosis using immune-related molecular subtypes. The use of machine learning (LASSO regression, Cox regression) enhances predictive modelling for clinical applications.

Introduction

Agree regarding the importance of monocytes and neutrophils as effectors of innate responses during sepsis – required for microbial clearance and recruiting other immune cells. CD4 T cells are important for modulation but I recommend mentioning the role of regulating immune responses through cells such as Tregs that are needed to suppress exaggerated cytokine responses. CD8 effectors will be important here too. Also dysfunction in these cells may lead to, for example, inappropriate interferon responses that help locally (paracrine signalling) as seen in severe COVID-19 infection.

I recommend that if the author mentions or is referencing a specific study (e.g., line 98) then please summarise the findings for the benefit of the reader before the critique.

Methods and results

The paper utilises a well-structured bioinformatics approach (WGCNA, GSVA, and immune infiltration analysis). Inclusion of an independent validation cohort strengthens the reliability of findings.

The training dataset was a combination of GSE65682 (where there was complete data) and new recruits. It would be good to include how sepsis was diagnosed (criteria) and these participants. Also, a demographic table to understand the cohort – as a training dataset the readers will want to know how relevant they were, was there a risk of bias? Were there differences in demographics or clinical characteristics between the training and validation sets? What was the mortality rates (given the eventual risk groups are based on progression to death or risk based on another prognosis tool?). A table summarising this would add value.

It would help the reader if there were a list of genes included used to associate with the different cell types and subsets – perhaps in a table.

The k-value selection for clustering is mentioned, but further justification would strengthen the argument. Why was k=2 chosen? Was this the optimum value (the text suggests this was not assessed)? How was cluster stability assessed?

In WGCNA, the soft thresholding power selection (R² maximisation) is briefly mentioned but should be explained further.

The study uses HR (hazard ratio) and confidence intervals (CI) for survival analysis. However, was a power calculation performed to ensure sample size sufficiency?

Were multiple testing corrections (e.g., Bonferroni, FDR) applied for differential expression and enrichment analyses?

In general, the bioinformatics workflow is well structured. Including: unsupervised clustering (ConsensusClusterPlus) to identify molecular subtypes, WGCNA (Weighted Gene Co-expression Network Analysis) used for gene module detection, Gene Ontology (GO) and KEGG pathway analyses for biological function, and immune infiltration analysis (CIBERSORT) for immune cell distributions.

The authors have stratified further by assessing the tool used for independent variables that will influence mortality such as age, but what about the the sepsis causing organism since this may affect pathogenesis and progression to severe sepsis?

Discussion

This is comprehensive and the authors have appreciated the study limitations and future directions.

6. PLOS authors have the option to publish the peer review history of their article (what does this mean? ). If published, this will include your full peer review and any attached files.

**Do you want your identity to be public for this peer review?** For information about this choice, including consent withdrawal, please see our Privacy Policy .

Reviewer #1: No

Reviewer #2: No

---

## [Author Response · Author response to Decision Letter 1]

27 Feb 2025

Dear Editor and Reviewers,

Thank you for your comment. We have revised our manuscript according to these comments. Responses to all the comments are as follows:

Response to Journal requirements

Comment 1. Please ensure that your manuscript meets PLOS ONE's style requirements, including those for file naming. The PLOS ONE style templates can be found at https://journals.plos.org/plosone/s/file?id=wjVg/PLOSOne_formatting_sample_main_body.pdf and https://journals.plos.org/plosone/s/file?id=ba62/PLOSOne_formatting_sample_title_authors_affiliations.pdf

Response 1: Yes, the manuscript has been prepared according to the style requirements.

Comment 2. Please note that PLOS ONE has specific guidelines on code sharing for submissions in which author-generated code underpins the findings in the manuscript. In these cases, we expect all author-generated code to be made available without restrictions upon publication of the work. Please review our guidelines at https://journals.plos.org/plosone/s/materials-and-software-sharing#loc-sharing-code and ensure that your code is shared in a way that follows best practice and facilitates reproducibility and reuse.

Response 2: Yes, we have uploaded all codes as supplementary materials.

Comment 3. For studies involving third-party data, we encourage authors to share any data specific to their analyses that they can legally distribute. PLOS recognizes, however, that authors may be using third-party data they do not have the rights to share. When third-party data cannot be publicly shared, authors must provide all information necessary for interested researchers to apply to gain access to the data. (https://journals.plos.org/plosone/s/data-availability#loc-acceptable-data-access-restrictions) For any third-party data that the authors cannot legally distribute, they should include the following information in their Data Availability Statement upon submission:1) A description of the data set and the third-party source. 2) If applicable, verification of permission to use the data set. 3) Confirmation of whether the authors received any special privileges in accessing the data that other researchers would not have. 4) All necessary contact information others would need to apply to gain access to the data

Response 3: Yes, we have updated these descriptions in the Data Availability Statement as follows: The expression levels of immune-related genes from sepsis patients can be available from the GEO without any special privileges (GSE65682: https://www.ncbi.nlm.nih.gov/geo/query/acc.cgi?acc=GSE65682), which is publicly for everyone. The validation dataset and original codes are within the manuscript and its Supporting Information files.

Comment 4: PLOS requires an ORCID iD for the corresponding author in Editorial Manager on papers submitted after December 6th, 2016. Please ensure that you have an ORCID iD and that it is validated in Editorial Manager. To do this, go to ‘Update my Information’ (in the upper left-hand corner of the main menu), and click on the Fetch/Validate link next to the ORCID field. This will take you to the ORCID site and allow you to create a new iD or authenticate a pre-existing iD in Editorial Manager.

Response 4: Yes, we have added the ORICID for corresponding author.

Comment 5. Please review your reference list to ensure that it is complete and correct. If you have cited papers that have been retracted, please include the rationale for doing so in the manuscript text or remove these references and replace them with relevant current references. Any changes to the reference list should be mentioned in the rebuttal letter that accompanies your revised manuscript. If you need to cite a retracted article, indicate the article’s retracted status in the References list and include a citation and full reference for the retraction notice.

Response 5: Yes, we have reviewed all reference lists and made sure they are right. The added references have been mentioned.

Response to Reviewer #1

This is a very complex study involving a considerable amount of work with clinically important objectives. The manuscript is well written, and the discussion is argued convincingly. The discussion of the pros and the limitations of the study were insightful. Comment 1: Could the authors consider adding further specific details about the findings of other similar prior studies in the discussion?

Response 1: Thank you for your comment. We have the following discussion and make some comparisons with a previous study in data usage and analysis methods, added analysis content, and the advantages and disadvantages of the model: A recent study also analyzed the associations between immune-related genes and clinical prognosis or the immune microenvironment in sepsis [17]. There are evident differences between our research and previous studies that we need to address. First, although we utilized the same dataset, our research diverges from previous studies in terms of data processing strategies. While prior studies divided the GEO dataset into training and validation sets for model construction, our study used the GEO dataset exclusively as a training set and incorporated an independently collected validation cohort. Secondly, our research performed an unsupervised clustering analysis on the GEO dataset, identifying two molecular subtypes with significantly distinct clinical prognoses, biological functions, and immunologic backgrounds, which had not been done in earlier studies. Furthermore, our model differs from those developed in prior research; whereas previous studies created a prognostic model comprising seven immune-related genes with a consistency index of 0.78, our model includes twenty-one immune-related genes and achieves a consistency index of 0.83. In terms of predictive ability, our model’s predictive ability was relatively high. Finally, our research established a personalized risk scoring system based on risk scores and estimated the mortality risk for patients at each critical time point. Additionally, we connected the molecular subtypes of patients with sepsis to the prognostic risk patterns, forming a “molecular subtype-risk identification-risk assessment” system. Our study not only enhances but also expands upon prior research, providing robust supporting data for validation.

We then added some other gene prognosis model studies: Some other gene prognosis models were also developed. Yang et al developed a nine-gene model and the AUCs were 0.83, 0.81, and 0.77 at 7, 14 and 21 days. But the internal and external validation datasets were lacking for this study.

17. Chen ZH, Zhang WY, Ye H, Guo YQ, Zhang K, Fang XM. A signature of immune-related genes correlating with clinical prognosis and immune microenvironment in sepsis. Bmc Bioinformatics. 2023;24(1):20. http://doi.org/10.1186/s12859-023-05134-1

31. Huang C, Liu Z, Guo Y, Wang W, Yuan Z, Guan Y, et al. scCancerExplorer: a comprehensive database for interactively exploring single-cell multi-omics data of human pan-cancer. Nucleic Acids Res. 2025;53(D1): D1526-35. http://doi.org/10.1093/nar/gkae1100

32. Peng Y, Wu Q, Ding X, Wang L, Gong H, Feng C, et al. A hypoxia- and lactate metabolism-related gene signature to predict prognosis of sepsis: discovery and validation in independent cohorts. Eur J Med Res. 2023;28(1):320. http://doi.org/10.1186/s40001-023-01307-z

33. Liang S, Xing M, Chen X, Peng J, Song Z, Zou W. Predicting the prognosis in patients with sepsis by a pyroptosis-related gene signature. Front Immunol. 2022; 13:1110602. http://doi.org/10.3389/fimmu.2022.1110602

(1) I have some minor questions

Comment 2: Could the authors expand in the discussion regarding any paradoxes that there may be in the biological function and immune status (immune cells and checkpoint genes) seen between cluster A (Figures 5B, C&D) and the high-risk groups (Fig 9C, D & E).

Response 2: Thank you for your comment. We have added the following discussion: In sepsis, both the function and the number of various immune cells undergo significant changes, leading to an immunosuppressive state. These alterations include shifts in cell counts, functional impairments, and imbalances in the secretion of pro-inflammatory and anti-inflammatory factors. Together, these disruptions contribute to immune system dysfunction, which adversely impacts the clinical outcomes of patients. This study found that patients in cluster A had a worse prognosis compared to those in cluster B. Immune infiltration analysis revealed that most immune cells and their functions were suppressed in cluster A patients, and the expression of immune checkpoint-related genes was relatively low, supporting the notion of immune system suppression in sepsis. Similar trends were observed in high-risk and low-risk groups. However, some immune cells and functions, such as iDCs, macrophages, and Tregs, were elevated in patients with a poor prognosis, particularly in cluster A. This may be linked to their specific roles: iDCs, although they have a strong capacity for antigen uptake, are functionally inactive and have a limited ability to activate T-cell responses; macrophages primarily regulate non-specific immunity and help maintain immune tolerance. Their increased presence may represent a stress response. Additionally, we observed increased expression of two immune checkpoint genes, CD200R1 and LAIR1, both of which are immunosuppressive. This upregulation is associated with the immunosuppressive effects seen in sepsis patients. In the high-risk group, LAIR1 was also elevated, as were immunosuppressive genes CD276 and PDCD1LG2, which is consistent with immunosuppressive status in sepsis patients.

Comment 3: Could the authors expand in the discussion their hypotheses regarding the poor prognostic performance of variables that have been associated with poor prognosis in prior phenotypic studies e.g., age, sex. Also, it may be interesting to expand on the poor performance of "endotype" in both the training and validation datasets.

Response 3: Thank you for your comment. We have added the following discussion: In multivariate analyses, we found confirmed that the risk score based on immune-related genes was an independent prognosis factor in both training and validation datasets. We also found that age was independently associated with poor prognosis in sepsis. Older adults are immunocompromised, making them more vulnerable to sepsis, and their prognosis tends to be poorer. We did not find the correlation of gender with prognosis in sepsis patients. Previous study suggested that gender may influence age-associated sepsis outcomes. We assume that there mab be interactions between age and gender, which need to be further confirmed [37]. We also found no association between endotype and prognosis in sepsis patients. Endotype is genotyped based on the genome, but sepsis is characterized by rapid disease progression, requiring quick clinical decision-making. Genotype-based endotyping may therefore have certain time lags. Additionally, the variation in patient characteristics could play a role. For instance, sepsis patients often present with immunosuppression, rapid disease progression, and the administration of specific treatments, all of which may influence the relationship between molecular typing and prognosis.

I have some minor suggestions re: visualization of the data:

Comment 4: Figure 2A - the labelling is not clearly visualized for a number of the genes in this figure, could the resolution of this figure be optimized? Could this be achieved by reducing the number of genes in the figure to only those with a cox p value of 0.05 or less? Could changing the colour of the negative correlations from pink also improve the resolution of correlations.

Response 4: Thank you for your comment. We have revised Figure 2A.

Comment 5: Maintaining consistency of the colors representing clusters A and B in all figures and graphs could be considered to help the reader easily identify the relevant cluster.

Response 5: Yes, we have checked all Figures and maintained the consistency of colors for cluster A and B. We marked the red for cluster A and blue for cluster B.

Comment 6: Is the unit for K-M (Fig 2F) intended to be days rather than years?

Response 6: Yes, the unit of K-M in Figure 2F should be days. We have revised this.

Response to Reviewer #2

Overall, an informative and interesting study. The study provides valuable insights into sepsis prognosis using immune-related molecular subtypes. The use of machine learning (LASSO regression, Cox regression) enhances predictive modelling for clinical applications.

Comment 1: Introduction: Agree regarding the importance of monocytes and neutrophils as effectors of innate responses during sepsis – required for microbial clearance and recruiting other immune cells. CD4 T cells are important for modulation but I recommend mentioning the role of regulating immune responses through cells such as Tregs that are needed to suppress exaggerated cytokine responses. CD8 effectors will be important here too. Also, dysfunction in these cells may lead to, for example, inappropriate interferon responses that help locally (paracrine signaling) as seen in severe COVID-19 infection. I recommend that if the author mentions or is referencing a specific study (e.g., line 98) then please summarize the findings for the benefit of the reader before the critique.

Response 1: Thank you for your comment. We have added the following descriptions: Previous studies have focused on the effect of individual genes on sepsis. Arunachalam et al. investigated the role of P2Y2 purinergic receptors in liver injury and sepsis. They found that mice lacking P2Y2 receptors exhibited reduced inflammation, less liver damage, and improved survival in response to both inflammatory liver injury and sepsis. These P2Y2 knockout mice also maintained normal serum arginine levels, preventing immune dysregulation and increased bacteremia seen in wild-type mice. The findings suggest that P2Y2 receptors play a crucial role in the pathophysiology of liver injury and sepsis, and targeting this receptor could potentially mitigate cytokine storms and improve outcomes in sepsis [14]. Liu et al. found that FCGR2C was the only gene differentially expressed between survivors and non-survivors in a cohort of 81 septic patients. FCGR2C was more predictive than the SOFA score in evaluating the prognosis of septic patients [15]. Jian et al. found that the level of UCP2 in blood cells of sepsis patients was significantly higher than in healthy controls, both at the mRNA and protein levels. UCP2 in blood cells may serve as a specific biomarker for sepsis, and its level is positively correlated with the severity of sepsis. [16]. These studies on individual genes have been insufficient in fully elucidating the complex pathophysiological processes of sepsis.

14. Arunachalam AR, Samuel SS, Mani A, Maynard JP, Stayer KM, Dybbro E, et al. P2Y2 purinergic receptor gene deletion protects mice from bacterial endotoxin and sepsis-associated liver injury and mortality. Am J Physiol-Gastr L. 2023;325(5):G471-91. http://doi.org/10.1152/ajpgi.00090.2023

15. Liu S, Zhang YL, Zhang LY, Zhao GJ, Lu ZQ. FCGR2C: An emerging immune gene for predicting sepsis outcome. Front Immunol. 2022;13:1028785. http://doi.org/10.3389/fimmu.2022.1028785

16. Jiang ZM, Yang QH, Zhu CQ. UCP2 in early diagnosis and prognosis of sepsis. Eur Rev Med Pharmaco. 2017;21(3):549-53.

Comment on Methods and results

Comment 2: The paper utilizes a well-structured bioinformatics approach (WGCNA, GSVA, and immune infiltration analysis). Inclusion of an independent validation cohort strengthens the reliability of findings. The training dataset was a combination of GSE65682 (where there was complete data) and new recruits. It would be good to include how sepsis was diagnosed (criteria) and these participants. Also, a demographic table to understand the cohort – as a training dataset the readers will want to know how relevant they were, was there a risk of bias? Were there differences in demographics or clinical characteristics between the training and validation sets? What was the mortality rates (given the ev

---

## [Decision Letter · Decision Letter 1]

PONE-D-24-57866R1Immune-associated molecular classification and prognosis signature of sepsisPLOS ONE

Dear Dr. Wang,

Thank you for submitting your manuscript to PLOS ONE. After careful consideration, we feel that it has merit but does not fully meet PLOS ONE’s publication criteria as it currently stands. Therefore, we invite you to submit a revised version of the manuscript that addresses the points raised during the review process.

We look forward to receiving your revised manuscript.

Kind regards,

Nishel Mohan Shah, PhD

Academic Editor

PLOS ONE

Journal Requirements:

Reviewers' comments:

Reviewer's Responses to Questions

**Comments to the Author**

1. If the authors have adequately addressed your comments raised in a previous round of review and you feel that this manuscript is now acceptable for publication, you may indicate that here to bypass the “Comments to the Author” section, enter your conflict of interest statement in the “Confidential to Editor” section, and submit your "Accept" recommendation.

Reviewer #1: (No Response)

Reviewer #2: (No Response)

2. Is the manuscript technically sound, and do the data support the conclusions?

Reviewer #1: Yes

Reviewer #2: Yes

3. Has the statistical analysis been performed appropriately and rigorously? 

Reviewer #1: Yes

Reviewer #2: Yes

4. Have the authors made all data underlying the findings in their manuscript fully available?

Reviewer #1: Yes

Reviewer #2: Yes

5. Is the manuscript presented in an intelligible fashion and written in standard English?

Reviewer #1: Yes

Reviewer #2: Yes

6. Review Comments to the Author

Reviewer #1: This is a very interesting and impactful study which merits publication. It has important objectives and the findings provide the basis for further investigation of sepsis molecular signatures, immune-related pathways and individualised prognosis for sepsis.

The points raised following the last review have been responded to thoughtfully.

Feedback in overview:

The impact of this study may be communicated more robustly following some minor revisions. The manuscript is written in standard English and overall, is very well written, particularly the abstract and the last part of the discussion. However, in some areas it has less clarity and is less fluent, not uncommon when a manuscript has been formulated by multiple co-authors. Given that manuscripts are not copyedited by PLOS ONE, I would recommend that the authors consider further self-edits to address this. Also, I recommend the addition of further relevant citations to support statements regarding prior knowledge e.g., in the introduction and discussion.

Specific feedback:

Introduction:

Line 70 - As this reference is of a study looking at neonatal infections, additional relevant references are recommended to support this statement.

Line 73 - An additional reference relating to findings in humans would be recommended here.

Line 107 - Regarding FCGR2C - I presume upregulation of this gene was seen in non-survivors but I would recommend clarification in the statement.

Line 112 - It would be helpful for the authors to expand re: what the study aims of the cited studies were at this point. This would help clarify the comparisons that the authors make with their study design. I note that they return to this later on in the manuscript, but feel that this would improve the fluency of the manuscript if the topic was also addressed prior to leading into the advantages of the authors' study design.

Line 152& 163/ Table 1 - This table is a valuable inclusion in the manuscript and the authors appropriately highlight the presence of missing data. I would recommend that they state clearly how they managed the missing data. What statistical analyses were performed to account for the missing data? Also, this Table might be better placed in the "Results" section.

Additionally, there is a high percentage of missing data in the training dataset e.g., unknown classification for thrombocytopenia in the training group etc., Can the authors discuss re: their confidence in the significance of the statistical differences seen in the parameters (between the training and validation sets) when a large portion of the categorisation is unknown.

I would also recommend expanding on this point in the discussion re: study limitations e.g., on how this could affect the results interpretation of the data (particularly the clinical prognostic tools).

Line 282 - Figure 1. I would recommend checking the table for typos (Kaplan-Meier) and consistency of content. The sample number appears differently in the table and in the text (890 - line 282; 802 - line 141).

Line 375 - Figures - 7A-7N - The unit on the K-M curves is years and is likely to be a typo.

Line 385 - Figure 7 - I would recommend changing the labelling on some of these graphs to aid the reader. The legend is well written. However, on the graphs, it would be recommended to change 7 E/F; I/J; M/N from yes/ no to the parameter being evaluated e.g., ICU-acquired infection or Non-ICU-acquired infection etc.

Discussion:

I would recommend including citations to support statements made.

I would recommend clarification of the sentences in Lines 448 & 512. They appear to potentially contradict each other.

I would also recommend clarification of the statement in Line 512 regarding immune checkpoint-related genes being relatively low. Often an increase in expression of checkpoint inhibitory markers (at the protein level) is associated with immune suppression. A suitable citation would be recommended here. Also, does this contradict statements made later e.g., Line 607-608?

I would also recommend that the authors include further relevant citations throughout the discussion to support statements e.g., line 512

E.g., Line 517-519 - I would recommend adding supportive citations.

Line 527 - This section is very interesting. The impact of this study would be enhanced by the authors contrasting the interpretation of the findings of the studies mentioned (with their study) and the potential implications.

Line 580 - I would recommend supporting this statement by adding the specific details from the study findings.

Line 584 - I would recommend clarification of the differences seen where it is stated that the expression is "abnormal".

Line 594-596 - This section would benefit from additional context here.

Line 600 - This is very good to include but may benefit from being included earlier in the discussion, before the earlier mention of immune checkpoint molecules.

Reviewer #2: I would like to thank the authors for addressing my comments and editing the manuscript accordingly.

7. PLOS authors have the option to publish the peer review history of their article (what does this mean? ). If published, this will include your full peer review and any attached files.

**Do you want your identity to be public for this peer review?** For information about this choice, including consent withdrawal, please see our Privacy Policy .

Reviewer #1: No

Reviewer #2: No

---

## [Author Response · Author response to Decision Letter 2]

25 Apr 2025

Response to Editorial

Comment: Journal Requirements: Please review your reference list to ensure that it is complete and correct. If you have cited papers that have been retracted, please include the rationale for doing so in the manuscript text or remove these references and replace them with relevant current references. Any changes to the reference list should be mentioned in the rebuttal letter that accompanies your revised manuscript. If you need to cite a retracted article, indicate the article’s retracted status in the References list and also include a citation and full reference for the retraction notice.

Response: Yes, we have checked all references one by one, and there are no retracted papers in the references lists.

Response to Reviewers

Reviewer #1: This is a very interesting and impactful study which merits publication. It has important objectives, and the findings provide the basis for further investigation of sepsis molecular signatures, immune-related pathways and individualized prognosis for sepsis. The points raised following the last review have been responded to thoughtfully. Feedback in overview: The impact of this study may be communicated more robustly following some minor revisions. The manuscript is written in standard English and overall, is very well written, particularly the abstract and the last part of the discussion. However, in some areas it has less clarity and is less fluent, not uncommon when a manuscript has been formulated by multiple co-authors. Given that manuscripts are not copyedited by PLOS ONE, I would recommend that the authors consider further self-edits to address this. Also, I recommend the addition of further relevant citations to support statements regarding prior knowledge e.g., in the introduction and discussion.

Introduction:

Comment 1: Line 70 - As this reference is of a study looking at neonatal infections, additional relevant references are recommended to support this statement.

Response 1: Yes, we have updated this reference citation using the following one:

3. Rudd KE, Johnson SC, Agesa KM, Shackelford KA, Tsoi D, Kievlan DR, et al. Global, regional, and national sepsis incidence and mortality, 1990-2017: analysis for the Global Burden of Disease Study. Lancet. 2020;395(10219):200-11. http://doi.org/10.1016/S0140-6736(19)32989-7

Comment 2: Line 73 - An additional reference relating to findings in humans would be recommended here.

Response 2: Yes, we have added the following reference:

6. Apitzsch S, Larsson L, Larsson AK, Linder A. The physical and mental impact of surviving sepsis - a qualitative study of experiences and perceptions among a Swedish sample. Arch Public Health. 2021;79(1):66. http://doi.org/10.1186/s13690-021-00585-

Comment 3: Line 107 - Regarding FCGR2C - I presume upregulation of this gene was seen in non-survivors but I would recommend clarification in the statement.

Response 3: Yes, we revised these descriptions as follows: Liu et al. found that FCGR2C was the only down-regulated gene differentially expressed between survivors and non-survivors in a cohort of 81 septic patients.

Comment 4: Line 112 - It would be helpful for the authors to expand re: what the study aims of the cited studies were at this point. This would help clarify the comparisons that the authors make with their study design. I note that they return to this later on in the manuscript but feel that this would improve the fluency of the manuscript if the topic was also addressed prior to leading into the advantages of the authors' study design.

Response 4: Thank you for your advice. What we would like to clarify here is that previous studies focused on individual genes were unable to fully explain the complex physiological and pathological processes of sepsis. Although there have been studies on immune gene sets and sepsis prognosis, our research differs from previous studies in several important aspects such as molecular subtypes, immune-cell infiltration, individual risk evaluation, and independent validation cohort data, which is a general description. Because it is impossible for us to directly present our specific results in the Introduction Section. Later in the Discussion Section of the article, we present the specific details of these differences.

Comment 5: Line 152& 163/ Table 1 - This table is a valuable inclusion in the manuscript and the authors appropriately highlight the presence of missing data. I would recommend that they state clearly how they managed the missing data. What statistical analyses were performed to account for the missing data? Also, this Table might be better placed in the "Results" section. Additionally, there is a high percentage of missing data in the training dataset e.g., unknown classification for thrombocytopenia in the training group etc., Can the authors discuss re: their confidence in the significance of the statistical differences seen in the parameters (between the training and validation sets) when a large portion of the categorization is unknown. I would also recommend expanding on this point in the discussion re: study limitations e.g., on how this could affect the results interpretation of the data (particularly the clinical prognostic tools).

Response 5: Thank you for your advice. Table 1 has been placed in the Results Section.

For missing data, we handle the missing data as the third type of variable to make full use of this data. We further performed the stratified analyses for these variables. We still found significant differences in survival outcomes between high-risk and low-risk group (revised Figure 7). We also added the following descriptions in the study limitations: Second, the variables (pneumonia, thrombocytopenia, infection type, and diabetes) had some missing data. We included these variables in the multivariate cox regression to make full use of this dataset. This may generate some information bias. The stratified analyses in the missing data and results from the validation dataset weaken the effects of missing data on results.

Comment 6: Line 282 - Figure 1. I would recommend checking the table for typos (Kaplan-Meier) and consistency of content. The sample number appears differently in the table and in the text (890 - line 282; 802 - line 141).

Response 6: Yes, we have revised this. It should be 802.

Comment 7: Line 375 - Figures - 7A-7N - The unit on the K-M curves is years and is likely to be a typo. Line 385 - Figure 7 - I would recommend changing the labelling on some of these graphs to aid the reader. The legend is well written. However, on the graphs, it would be recommended to change 7 E/F; I/J; M/N from yes/ no to the parameter being evaluated e.g., ICU-acquired infection or Non-ICU-acquired infection etc.

Response 7: Yes, we have revised the unit in Figure 7. It should be Days. The labels were also revised according to the figure legends.

Comment 8: Discussion: I would recommend including citations to support statements made. I would recommend clarification of the sentences in Lines 448 & 512. They appear to potentially contradict each other.

Response 8: Yes, we revised this description as follows: Immune infiltration analysis showed that Cluster B had higher immune cell counts (CD8+ T cells, NK cells) than Cluster A, and immune function levels were also higher in Cluster B than in Cluster A.

Comment 9: I would also recommend clarification of the statement in Line 512 regarding immune checkpoint-related genes being relatively low. Often an increase in expression of checkpoint inhibitory markers (at the protein level) is associated with immune suppression. A suitable citation would be recommended here. Also, does this contradict statements made later e.g., Line 607-608?

Response 9: Yes, for line 512, we revised this description as follows: Immune infiltration analysis revealed that most immune cells and their functions were suppressed in cluster A patients, and the expression of immune checkpoint inhibitory genes was relatively elevated, supporting the notion of immune system suppression in sepsis. The following reference was cited.

36. Liu YC, Shou ST, Chai YF. Immune checkpoints in sepsis: New hopes and challenges. Int Rev Immunol. 2022;41(2):207-16. http://doi.org/10.1080/08830185.2021.1884247

Comment 10: I would also recommend that the authors include further relevant citations throughout the discussion to support statements e.g., line 512 E.g., Line 517-519 - I would recommend adding supportive citations.

Response 10: Yes, the following references were cited:

37.Guermonprez P, Valladeau J, Zitvogel L, Thery C, Amigorena S. Antigen presentation and T cell stimulation by dendritic cells. Annu Rev Immunol. 2002;20:621-67. http://doi.org/10.1146/annurev.immunol.20.100301.064828

38. Luo B, Gan W, Liu Z, Shen Z, Wang J, Shi R, et al. Erythropoeitin Signaling in Macrophages Promotes Dying Cell Clearance and Immune Tolerance. Immunity. 2016;44(2):287-302. http://doi.org/10.1016/j.immuni.2016.01.002

Comment 11: Line 527 - This section is very interesting. The impact of this study would be enhanced by the authors contrasting the interpretation of the findings of the studies mentioned (with their study) and the potential implications.

Response 11: Thank you for your comment. We added the following descriptions as follows: In our study, we also obtained two subtypes (cluster A and cluster B) based on immune-related genes, and the cluster A had poorer prognosis than cluster B. Our study provided a different perspective for sepsis subtypes.

Comment 12: Line 580 - I would recommend supporting this statement by adding the specific details from the study findings.

Response 12: Yes, these statements have been revised as follows: we observed that the levels of Th1 and Th2 cells were decreased in high-risk group.

Comment 13: Line 584 - I would recommend clarification of the differences seen where it is stated that the expression is "abnormal".

Response 13: Yes, we have revised these descriptions using “increased” or “decreased” instead of “abnormal”

Comment 14: Line 594-596 - This section would benefit from additional context here.

Response 14: A previous study investigated the leukocyte functions before and after passive blocking of PD-1 and PD-L1 with antibodies. White blood cells were derived from patients with sepsis, critically ill patients without sepsis and healthy controls. They

found that neutrophil and monocyte functions in patients with sepsis are decreased, which is related to CD8+ T cell- and NK cell-specific PD-1 and PD-L1 expression [50]. Interestingly, decreased CD8+ T cell effector function (such as IFNγ production capacity) was associated with elevated PD-L1 expression in neutrophils. After the application of anti-PD-1 or PD-L1 antibodies, the function of neutrophils and monocytes was restored, leukocyte apoptosis was reduced in patients with sepsis, and the function of T cell effects was also significantly restored, manifested by a significant increase in IFN-γ and IL-2 production [51]. This provides evidence for T cell exhaustion in patients with sepsis, but it is reversible.

Comment 15: Line 600 - This is very good to include but may benefit from being included earlier in the discussion, before the earlier mention of immune checkpoint molecules.

Response 15: Yes, we moved these descriptions in earlier section in the discussion.

---

## [Editor Report · Decision Letter 2]

Immune-associated molecular classification and prognosis signature of sepsis

PONE-D-24-57866R2

Dear Dr. Wang,

We’re pleased to inform you that your manuscript has been judged scientifically suitable for publication and will be formally accepted for publication once it meets all outstanding technical requirements.

Kind regards,

Nishel Mohan Shah, PhD

Academic Editor

PLOS ONE

Additional Editor Comments (optional):

Thank you for addressing the comments by our reviewer as well as the minor edits.
---

## [Editor Report · Acceptance letter]

PONE-D-24-57866R2

PLOS ONE

Dear Dr. Wang,

I'm pleased to inform you that your manuscript has been deemed suitable for publication in PLOS ONE. Congratulations! Your manuscript is now being handed over to our production team.

Kind regards,

on behalf of

Dr. Nishel Mohan Shah

Academic Editor

PLOS ONE